# BRCA2 binding through a cryptic repeated motif to HSF2BP oligomers does not impact meiotic recombination

Rania Ghouil[1,10], Simona Miron[1,10], Lieke Koornneef [2,10], Jasper Veerman [3], Maarten W. Paul [3], Marie-Hélène Le Du[1], Esther Sleddens-Linkels[4], Sari E. van Rossum-Fikkert[3,5], Yvette van Loon[3], Natalia Felipe-Medina[6], Alberto M. Pendas [6], Alex Maas[7], Jeroen Essers[3,5,8], Pierre Legrand [9], Willy M. Baarends [4], Roland Kanaar[3✉], Sophie Zinn-Justin[1✉] & Alex N. Zelensky [3✉]

BRCA2 and its interactors are required for meiotic homologous recombination (HR) and fertility. Loss of HSF2BP, a BRCA2 interactor, disrupts HR during spermatogenesis. We test the model postulating that HSF2BP localizes BRCA2 to meiotic HR sites, by solving the crystal structure of the BRCA2 fragment in complex with dimeric armadillo domain (ARM) of HSF2BP and disrupting this interaction in a mouse model. This reveals a repeated 23 amino acid motif in BRCA2, each binding the same conserved surface of one ARM domain. In the complex, two BRCA2 fragments hold together two ARM dimers, through a large interface responsible for the nanomolar affinity — the strongest interaction involving BRCA2 measured so far. Deleting exon 12, encoding the first repeat, from *mBrca2* disrupts BRCA2 binding to HSF2BP, but does not phenocopy HSF2BP loss. Thus, results herein suggest that the high-affinity oligomerization-inducing BRCA2-HSF2BP interaction is not required for RAD51 and DMC1 recombinase localization in meiotic HR.

[1] Institute for Integrative Biology of the Cell (I2BC), CEA, CNRS, Uni Paris-Sud, Uni Paris-Saclay, Gif-sur-Yvette, France. [2] Department of Developmental Biology, Oncode Institute, Erasmus University Medical Center, 3000 CA Rotterdam, The Netherlands. [3] Department of Molecular Genetics, Oncode Institute, Erasmus MC Cancer Institute, Erasmus University Medical Center, 3000 CA Rotterdam, The Netherlands. [4] Department of Developmental Biology, Erasmus University Medical Center, 3000 CA Rotterdam, The Netherlands. [5] Department of Radiation Oncology, Erasmus University Medical Center, 3000 CA Rotterdam, The Netherlands. [6] Molecular Mechanisms Program, Centro de Investigación del Cáncer and Instituto de Biología Molecular y Celular del Cáncer (CSIC-Universidad de Salamanca), Salamanca, Spain. [7] Department of Cell Biology, Erasmus University Medical Center, 3000 CA Rotterdam, The Netherlands. [8] Department of Vascular Surgery, Erasmus University Medical Center, 3000 CA Rotterdam, The Netherlands. [9] Synchrotron SOLEIL, L'Orme des Merisiers, Gif-sur-Yvette, France. [10] These authors contributed equally: Rania Ghouil, Simona Miron, Lieke Koornneef. ✉email: r.kanaar@erasmusmc.nl; Sophie.ZINN@cea.fr; a.zelensky@erasmusmc.nl

Homologous recombination (HR) is involved in many aspects of eukaryotic DNA metabolism and is indispensable in two contexts: resolving replication problems and in meiosis. Homology search and strand exchange, the key events in HR, are performed by a nucleoprotein filament formed by the strand exchange protein RAD51 assembled onto the 3′ single-stranded (ss) DNA overhang. In somatic animal cells, RAD51 loading onto ssDNA depends on BRCA2, which has multiple RAD51-binding sites and is required for focal accumulation of RAD51 at the sites of damage. Biochemical experiments suggest that BRCA2 acts as an HR mediator, displacing RPA, the protein that protects ssDNA by strongly binding to it, and forming functional RAD51 filament in its place[1]. In vitro, BRCA2 can perform this function autonomously, but in cells, it depends on its "partner and localizer" PALB2[2].

Although BRCA2 has been mostly studied in the context of HR in somatic cells, it arguably has a more prominent role in meiotic HR, as across a broad range of species, fertility defects are the most common consequence of BRCA2 loss[3–11]. In meiosis, HR functions to diversify as well as to preserve genetic information. This role is achieved by extending the core HR machinery (RAD51, BRCA2, and PALB2) with a set of meiosis-specific proteins, such as the DMC1 recombinase and the ssDNA-binding proteins MEIOB and SPATA22. BRCA2 binds DMC1 via the RAD51-binding BRC repeats encoded by BRCA2 exon 11[12–14] and a DMC1-specific site encoded by exon 14[13].

We identified HSF2BP as another BRCA2-binding protein, endogenously expressed in meiotic cells, and ectopically produced in cancer cells[15–17]. HSF2BP is required for meiotic HR during spermatogenesis, but in somatic cells, it inhibits HR during DNA interstrand crosslink repair by triggering BRCA2 degradation. In addition to BRCA2, HSF2BP has been reported to interact with transcription factors HSF2[18] and BNC1[19], both required for normal fertility in mice[20,21]. More recently, five groups independently reported that HSF2BP interacts with an uncharacterized protein named C19orf57, 4930432K21Rik, BRME1, MEIOKE21, or MAMERR. The two proteins co-localize in meiocytes, loss of BRME1 closely phenocopies loss of HSF2BP, and the two proteins can affect each other's stability[22–26]. The model put forward to explain the meiotic defects in knock out mice for either Hsf2bp or Brme1 follows the PALB2 paradigm: HSF2BP and BRME1 are proposed to act as "meiotic localizers" for BRCA2—and for each other[25,27]. However, how HSF2BP, BRME1, and PALB2 contribute to BRCA2 localization in meiotic HR remains to be established. Also, in contrast to complete dependence of BRCA2 on PALB2 in somatic HR, the loss of the two meiotic localizers (HSF2BP and BRME1) causes a milder meiotic phenotype than Brca2 deficiency in mice. Hsf2bp and even more so Brme1-knockout mouse models show pronounced sexual dimorphism: female meiotic defects are either weak[22,27] or not detected[16,23–26], while Brca2 deficiency affects both sexes[4].

HSF2BP contains an N-terminal α-helical oligomerisation domain and a C-terminal domain predicted to adopt an Armadillo fold[16,17,25]. We mapped the HSF2BP-BRCA2 interaction to the Armadillo domain of HSF2BP, and a 68 amino acid (aa) region of BRCA2 mostly encoded by exons 12 and 13, which is predicted to be disordered[16]. BRCA2 is a protein of 3418 aa that possesses a unique globular domain of 700 aa binding to ssDNA and the acidic protein DSS1[28]. The high disorder propensity of BRCA2 is proposed to ensure its structural plasticity and its ability to orchestrate complex molecular transactions while balancing multiple interactions[29,30]. However, interactions involving intrinsically disordered regions are difficult to predict and characterize structurally. Out of more than a dozen mapped protein interaction regions within the disordered part of BRCA2[31,32], only three were crystallized when bound to their folded partner,

either Rad51[33], PALB2[34] or PLK1[35]. In all cases, the BRCA2 fragments became folded upon transient interactions characterized by affinities on the micromolar range.

In this study, we provide a detailed biophysical characterization of the HSF2BP-BRCA2 interaction and the changes in oligomeric state it induces. Its low-nanomolar affinity is orders of magnitude stronger than any other measured interaction involving BRCA2. We also describe the 3D structure of the complex between HSF2BP and BRCA2, which confirms the predicted ARM fold of HSF2BP and reveals the existence of a cryptic repeated motif encoded by exons 12–13 of BRCA2, responsible for binding to ARM oligomers. Finally, results from a mouse line engineered to disrupt the binding suggest that contrary to the prediction of the "meiotic localizer" model, this evolutionarily conserved high-affinity oligomerisation-inducing interaction of BRCA2 with HSF2BP is not required for meiotic HR.

## Results

**HSF2BP ARM binds disordered BRCA2 peptide with high affinity.** Our previous analyses using coimmunoprecipitation identified the C-terminal part of HSF2BP I93-V334 and the BRCA2 fragment G2270-T2337 (F9) as interacting regions[16]. Extending this approach (Fig. 1a–c, Supplementary Fig. 1a, b), we narrowed down the minimal interaction region to E122-V334 in HSF2BP (fragment H3, hereafter referred as ARM, for "armadillo domain") and to N2288-T2337 in BRCA2 (fragment F15). Further truncations resulted in loss or reduction in co-precipitation efficiency. We also extended our site-directed mutagenesis mapping: using a homology model of the HSF2BP ARM domain, we predicted solvent-exposed structural neighbors of R200, which we previously found to be required for BRCA2 binding, and made substitutions based on human polymorphism data (dbSNP). Whereas in our initial blind screen[16] most substitutions preserved binding, GFP-HSF2BP mutated at residues N192, G199, Y238, N239 or N243 failed to co-precipitate Flag-F9 (Supplementary Fig. 1b, c). On the BRCA2 side, wild-type GFP-HSF2BP also did not interact with several Flag-F9 variants, mutated at residues S2309, R2318, or P2329 (Supplementary Fig. 1d). To characterize the direct interaction between HSF2BP and BRCA2 in vitro, we purified HSF2BP and its truncated variants, as well as the large BRCA2 fragment S2213-Q2342 (F0) showing high conservation during evolution (and including F15; Supplementary Fig. 2). We first performed an NMR analysis of F0, in order to identify the residues binding to ARM. Assignment and further analysis of the NMR Hn, N, Cα, Cβ and Co chemical shifts of $^{15}$N-, $^{13}$C-labeled BRCA2-F0 showed that this peptide is disordered in solution: it only forms transient α-helices; in particular, region N2291-S2303 folds into an α-helix that is present at about 25% (Supplementary Fig. 3). Addition of unlabeled ARM causes a global decrease of the intensities of the 2D NMR $^1$H-$^{15}$N HSQC peaks of $^{15}$N-labeled F0, with region S2252-Q2342 (further called $F_{NMR}$), including the transient α-helix, showing the largest decrease (intensity ratio lower than 0.4; Fig. 1d). We concluded that the chemical environment of this region, larger than the previously identified F15 fragment N2288-T2337, is significantly modified in the presence of the ARM domain. Isothermal Titration Calorimetry (ITC) experiments revealed that F0 binds to both HSF2BP and ARM with a nanomolar affinity and a stoichiometry of 0.5, i.e., one BRCA2 peptide binds to two HSF2BP/ARM (Fig. 1e; Table 1). These affinity and stoichiometry are consistent with the more than twofold decrease in intensity observed by NMR when adding one ARM to one $^{15}$N-labeled F0; indeed, in these conditions, half of the F0 molecules were free and half bound to ARM. Also, HFS2BP binds about 25-fold tighter to F0 than does ARM. As a control, we verified that HSF2BP mutant

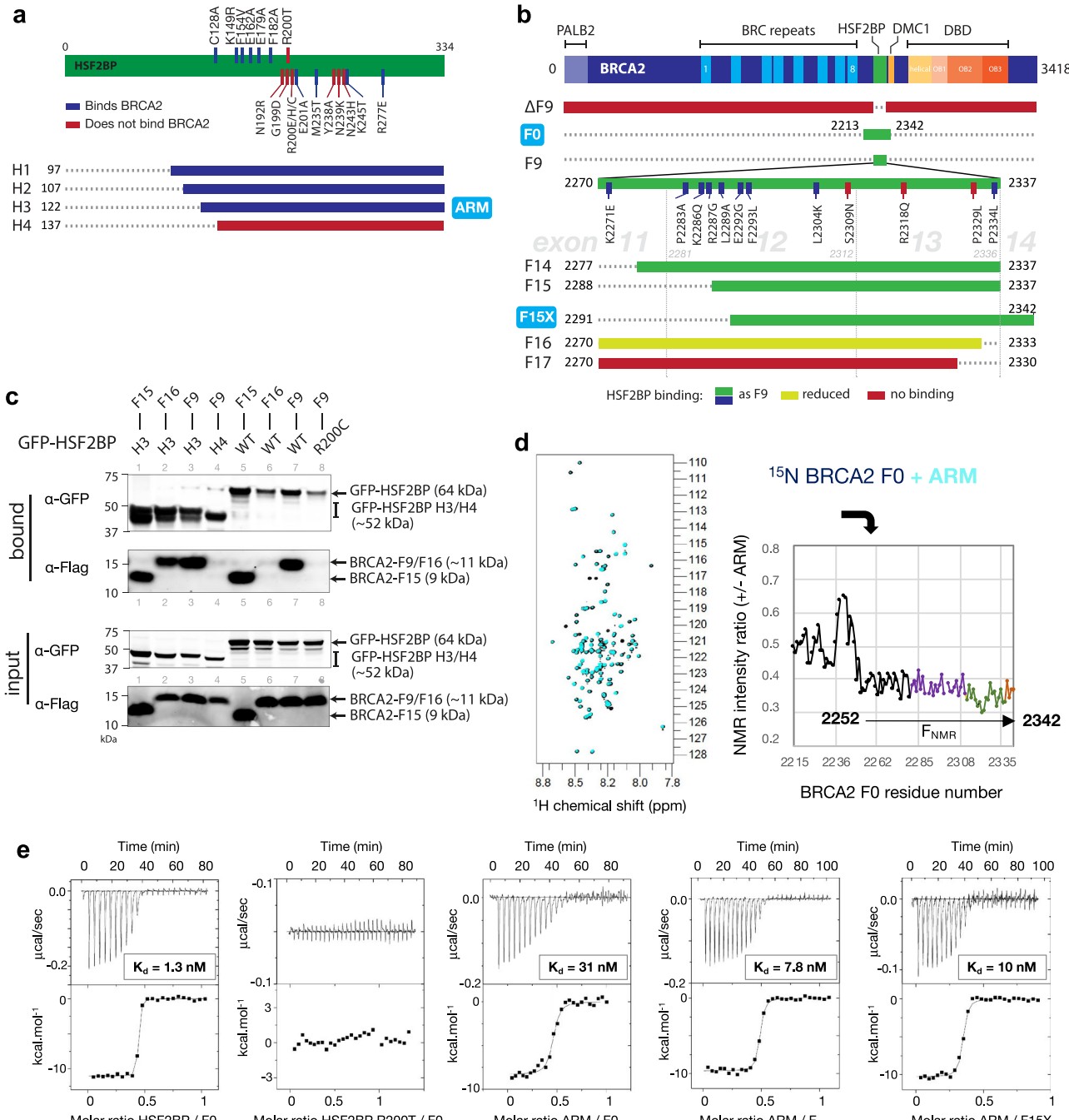

**Fig. 1 The ARM domain of HSF2BP binds with a nanomolar affinity to a 52 aa BRCA2 peptide. a** Schematic depiction of the truncation and substitution variants of HSF2BP used in this study. Substitutions tested previously are mapped above the bar, whereas those tested in this study are indicated below the bar. Truncation variants are colored based on their ability to bind BRCA2 peptides. **b** Schematic depiction of BRCA2 fragments and variants used in the study. Full-length BRCA2 is shown at the top with key domains indicated. Location of the fragment F9 identified previously and its truncations tested here are shown, with colors indicating the ability to bind HSF2BP. Fragments produced as recombinant proteins are indicated with blue labels. **c** Coimmunoprecipitation of GFP-HSF2BP (full-length wild-type (WT) or R200C variant, and fragments H3 (ARM) and H4) and indicated Flag-tagged BRCA2 variants. Proteins were transiently produced in HEK293T cells. **d** NMR characterization of BRCA2 residues involved in binding to the Armadillo domain of HSF2BP (ARM). 2D $^{1}$H-$^{15}$N HSQC spectra were recorded at 950 MHz and 283 K on the $^{15}$N-labeled BRCA2 fragment F0 (S2213-Q2342), either free (100 μM; dark blue) or in the presence of the unlabeled ARM domain (1:1 ratio; cyan). Ratios of peak intensities in the two conditions revealed that a set of peaks, corresponding to BRCA2 fragment $F_{NMR}$ (S2252-Q2342), decreased by more than 60% after the addition of ARM. The points and curve fragments in purple, green, and brown correspond to residues encoded by exon 12, 13, and 14 of BRCA2, respectively. **e** ITC curves that reveal how either HSF2BP or its ARM domain (in the instrument cell) interacts with the BRCA2 fragment F0, $F_{NMR}$, and F15X (N2291-Q2342) (in the instrument syringe). The dissociation constants ($K_d$) are indicated. All experiments were duplicated, and the dissociation constants, stoichiometry, and thermodynamics parameters of each experiment are recapitulated in Table 1.

**Table 1 Isothermal titration calorimetry data.**

|  | Kd (±error) (M) | n | ΔH (±error) (kcal/mol) | ΔG (kcal/mol) | −TΔS (kcal/mol) |
|---|---|---|---|---|---|
| HSF2BP FL vs. F0— expt 1 | 1.3E−09 (0.3) | 0.43 | −11.0 (0.1) | −11.8 | −0.8 |
| HSF2BP FL vs. F0—expt 2 | 1.3E−09 (0.3) | 0.47 | −10.9 (0.1) | −11.9 | −1.0 |
| HSF2BP FL vs. F15XΔ12—expt 1 | 1.5E−06 (0.7) | 0.89 | −1.7 (0.3) | −7.8 | −6.1 |
| HSF2BP FL vs. F15XΔ12—expt 2 | 2.1E−06 (0.9) | 0.90 | −1.8 (0.2) | −7.6 | −5.8 |
| ARM vs. F0—expt 1 | 31E−09 (9.0) | 0.45 | −8.3 (0.2) | −10 | −1.7 |
| ARM vs. F0—expt 2 | 25E−09 (7.0) | 0.49 | −11.9 (0.2) | −10.1 | 1.8 |
| ARM vs. $F_{NMR}$—expt 1 | 7.8E−09 (1.3) | 0.47 | −9.6 (0.1) | −10.8 | −1.2 |
| ARM vs. $F_{NMR}$—expt 2 | 13E−09 (4.0) | 0.51 | −9.4 (0.2) | −10.5 | −1.1 |
| ARM vs. F15X—expt 1 | 10E−09 (2.1) | 0.33 | −10.2 (0.1) | −10.6 | −0.4 |
| ARM vs. F15X—expt 2 | 9.4E−09 (2.3) | 0.40 | −9.2 (0.1) | −10.7 | −1.5 |

R200T does not bind to F0, consistently with our previous report[16]. We further compared the affinity of ARM for F0, $F_{NMR}$, and F15X (the recombinant peptide N2291-Q2342, similar to F15 N2288-T2337 used in cellular assays). The affinities of ARM for $F_{NMR}$ and F15X are not significantly different, being around 10 nM (Fig. 1e; Table 1). Unexpectedly, they are about threefold higher than the affinity measured between ARM and F0 (Fig. 1e; Table 1). Therefore, we decided to continue by focusing on the complex between ARM and F15X.

**HSF2BP ARM domain tetramerizes upon binding to BRCA2.** First, we characterized the molecular mass of the ARM domain either free or bound to F15X. Biophysical analysis by SEC-multi angle light scattering (MALS) (Size exclusion chromatography—multiple angle light scattering) and SEC-SAXS (Small-angle X-ray scattering) revealed that, if free ARM is dimeric, the complex is tetrameric with an estimated molecular weight of 94 kDa (SEC-MALS; Fig. 2a) or 109 kDa (SEC-SAXS; Fig. 2b), for a theoretical mass of four ARM bound to two F15X of 108 kDa. In parallel, intensity curves measured by SAXS on the complex gave a distance distribution reflecting a nearly spherical shape, with a Dmax of 104 Å and a Rg of 34 Å (Fig. 2b).

Crystals of the complex were obtained within a few days by hanging-drop vapor diffusion and diffracted up to 2.6 Å on PROXIMA-1 and PROXIMA-2 beamlines at the SOLEIL synchrotron. The structure of the complex was solved using a combination of Molecular Replacement and SAD approaches (see details in "Methods" section, Table 2 and Supplementary Fig. 4). The overall conformation of the structure is consistent with the SAXS data obtained in solution, as reflected by the low chi$^2$ value of 1.8 Å$^2$ obtained when fitting the SAXS curve deduced from the experimental structure to the experimental SAXS curve (Fig. 2b).

The crystal structure includes four ARM domains and two BRCA2 F15X peptides (Fig. 2c, d). The ARM domains A and D, as well as B and C, dimerize through a symmetric interface of about 950 Å$^2$, formed by their N-terminal regions from E122 to I156 (Fig. 2d; Supplementary Fig. 5a). This interface is mediated by hydrophobic residues from helices α1, helices α2, and the N-terminus of helices α3. In contrast, the ARM domains A and C, as well as B and D, have a very small direct interface of less than 100 Å$^2$. They are juxtaposed, one chain being rotated around its main axis by about 90° relatively to the other and interact mainly through BRCA2 (Fig. 2c). The BRCA2 peptide in orange (chain E) runs along the V-shaped groove formed by chains A and C. Similarly, the other BRCA2 peptide (in red; chain F) runs along the groove formed by chains B and D. At the center of the tetramer, a symmetric interface of about 250 Å$^2$ is observed between the ARM domains A and B, which involves helices α2 and helices α5 (Supplementary Fig. 5b). This interface is poorly conserved. In summary, two ARM dimers interact through two

BRCA2 peptides to form a tetramer; within the tetramer, two types of conserved interfaces are observed, either between monomers from the same dimer (chains A and D, as well as B and C, see boxed view in Fig. 2d), or between the ARM domains and the BRCA2 peptides (see main panel in Fig. 2d).

**Repeated motifs in BRCA2 hold together two HSF2BP ARM dimers.** The 3D structures of the complexes between, on the one hand, the ARM domains A and C and the peptide E, and on the other hand the ARM domains B and D and the peptide F, are remarkably similar (Fig. 3a). In these structures, two ARM monomers form a BRCA2-binding surface of 2740 Å$^2$, which is in the upper range of interaction surfaces, even for a complex between a folded domain and an intrinsically disordered peptide[36]. The BRCA2 peptide engages 48 aa in this interaction. In chains E and F, the N-terminal sequence, from N2291 to E2328, interacts with ARM domains C and D, respectively, whereas the C-terminal sequence, from D2312 to T2338, interacts with ARM domains A and B, respectively. The 3D structures of the ARM domains interacting with the same region of the peptide are highly similar, whereas the 3D structures of two ARM domains interacting with different regions of the BRCA2 peptide show some local structural variations, as measured by the root-mean-square deviations between their Cα atoms (see Table in Fig. 3a and Supplementary Fig. 6). Another remarkable feature of this complex is that similar surfaces of the ARM domains recognize the N-terminal and the C-terminal regions of the BRCA2 peptide (Fig. 3b; Supplementary Fig. 7). Indeed, a surface of about 1540 Å$^2$ formed by helix α1, helix α4 and the N-terminal region of α5, helix α7 and the N-terminal region of α8, helix α10 and loop α10α11, and finally loop α12α13 on one ARM domain interacts with the N-terminal sequence of the BRCA2 peptide. A smaller surface of 1200 Å$^2$ formed by helix α1, helix α4 and the N-terminal region of α5, helix α7 and the N-terminal region of α8, only the C-terminal region of α10 and loop α10α11 on the other ARM domain interacts with the C-terminal sequence of the peptide. The surface common to the two binding interfaces is conserved through evolution and mainly positively charged (Fig. 3c). The surface specific to the interface with the N-terminal region of the peptide, including the N-terminus of helices α7 and α10 and loop α12α13, is less conserved.

Further analysis of the interface revealed that, even more strikingly, the N-terminal and C-terminal sequences of the BRCA2 peptide interacting with different ARM domains have similar structures (Fig. 3d). BRCA2 motif 1, from N2291 to G2313, and BRCA2 motif 2, from T2314 to R2336, can be nicely superimposed, and interact with the same surface of their respective ARM domains. Sequence alignment between the two motifs identified five identical residues: D2294/2317, R2295/2318, S2303/2326 (also a proline in some organisms), L2304/2327, and

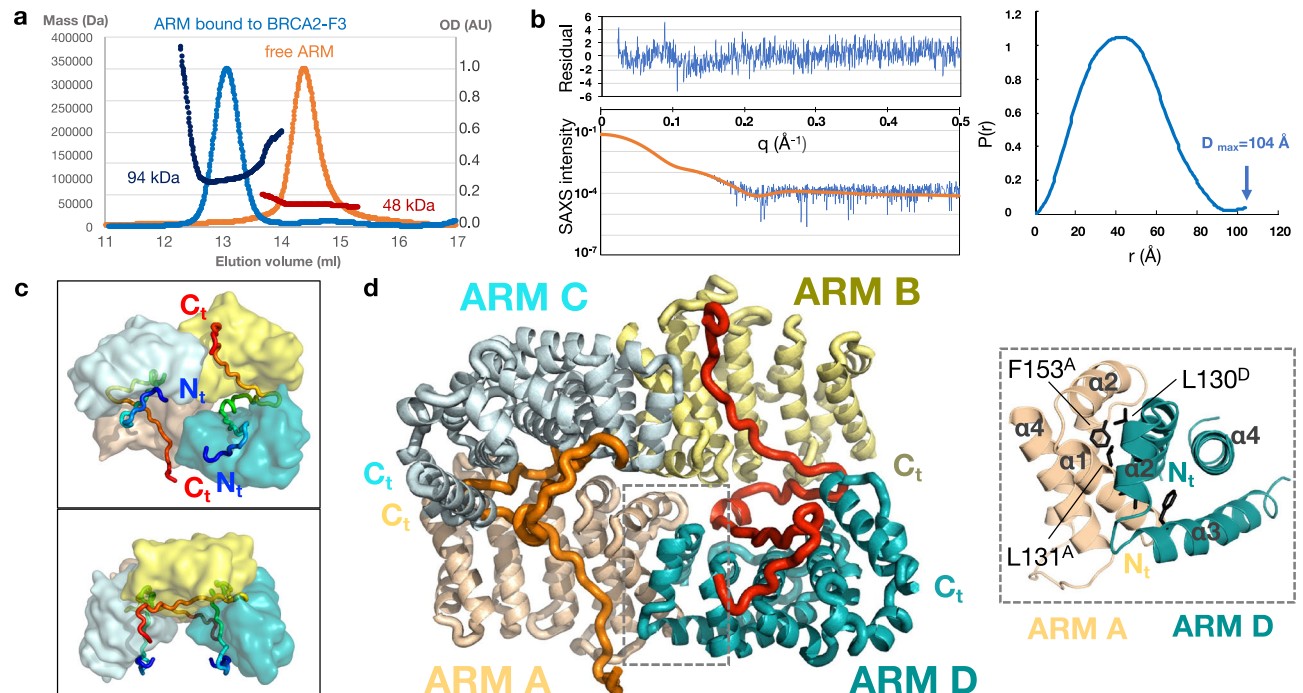

**Fig. 2 The ARM domain tetramerizes upon BRCA2 binding. a** SEC-MALS profiles from two independent experiments, performed either on free ARM (orange: OD normalized to 1; red: mass) or on ARM bound to F15X (light blue: OD normalized to 1; dark blue: mass) (column: Superdex 200 10/300 GL). **b** SEC-SAXS curve and resulting distance distribution obtained on ARM bound to F15X (blue). The experimental SAXS curve is compared to the theoretical SAXS curve calculated with CRYSOL from the X-ray structure of the complex (orange). Residual errors are plotted as a function of the scattering angle (resulting chi$^2$ value: 1.8 Å$^2$). **c, d** Different views of the crystal structure of the complex, illustrating how the ARM dimers, formed by chains A (wheat) and D (teal) and chains B (yellow) and C (pale blue), are held together through their interactions with the BRCA2 peptides. **c** The ARM domains are represented as surfaces, whereas the BRCA2 F15X peptides are displayed as tubes colored from their N-terminus (blue) to their C-terminus (red). **d** The ARM chains are represented as cartoons, and the BRCA2 F15X peptides as orange (chain E) and red (chain F) tubes. A zoom view of the dimerization interface between ARM chains A and D is displayed in a dotted box: only helices α1–α4 are displayed for clarity. The dimerization interface is mediated by hydrophobic residues from helices α1 (M123, A126, A127, L130, L131, and V134), helices α2 (V140, I144), and the N-terminus of helices α3 (L151, F153, and I156). About a quarter of this interface is due to the interaction between the highly conserved L131 and F153 from one monomer and the highly conserved L130 from the other monomer (side chains displayed as black sticks, L131 and F153 from chain A, as well as L130 from chain D are labeled).

P2311/2334, and two similar residues: A2306/P2329, S2309/ C2332 (T2332 in our construct), which are all conserved through evolution (Fig. 3e; Supplementary Fig. 2). These residues interact with conserved residues from the ARM domains, as indicated in Fig. 3e. For example, D2294/2317 contacts R200 (helix α5) through a set of hydrogen bonds/salt bridges; R2295/2318 is hydrogen-bonded to S281 (helix α10); L2304/2327 contacts I242 (helix α7); S2309/C2332 and P2311/2334 interact W132 (helix α1); S2309/C2332 also contacts G188 and is hydrogen-bonded to N192 (helix α4). These conserved interactions between motif 1 and motif 2 of BRCA2 and two ARM monomers belonging to two different dimers trigger tetramerization of the ARM domain. In summary, analysis of the crystal structure of the BRCA2-HSF2BP complex identified a repeated motif in BRCA2 that is able to bind to the ARM domain of HSF2BP, thus causing tetramerization of the dimeric ARM domain.

**Residues essential for HSF2BP binding to BRCA2.** We compared the interface observed in our crystal structure with the point mutants we designed based on homology modeling of the solvent-exposed structural neighbors of R200 and analyzed by coimmunoprecipitation (Fig. 1a–c). Substitutions N192R, R200E/ H/C, Y238A, N239K affected residues that are hydrogen-bonded to both motifs 1 and 2. Consistently, they totally abolished binding (Fig. 4a; Supplementary Fig. 1c, d). R277E modified a residue that is hydrogen-bonded only to motif 1; it strongly

decreased binding. G199D changed a residue that is completely buried in ARM, and abolished binding. N243H affected a residue that is totally buried at the interface with both motifs; it also totally abolished binding. M235T modified a residue that is half buried at the interface with both motifs. This variant still weakly bound to BRCA2. Only E201A and K245T, which changed two residues forming an intramolecular salt-bridge partially buried at the interface with both motifs, did not impact BRCA2 binding (Fig. 4a; Supplementary Fig. 1c, d). Altogether, coimmunoprecipitation assays validated the essential role played by the conserved ARM surface in binding to BRCA2 in human cells. Furthermore, these assays highlighted the critical role played by asparagine residues N192 and N239 from the ARM surface. Despite the lack of secondary structure elements in bound BRCA2, the presence of asparagine residues that interact through their side chains with the backbone amide proton and oxygen of BRCA2 residues mimics the formation of a β-sheet hydrogen bond network between the ARM domains and BRCA2 (Fig. 4b). These interactions are independent of the BRCA2 sequence, as they involve only the backbone atoms of BRCA2. On the BRCA2 side, consistently with our ITC results, the only mutations clearly decreasing or abolishing the interaction are in the region L2304-P2329 (Fig. 1b). Most mutations of residues defining motif 1 (L2304K, S2309N), and motif 2 (R2318Q, P2329L) either strongly decreased or abolished binding to HSF2BP (Fig. 4c, d; Supplementary Fig. 1e). In contrast, mutations of the highly conserved BRCA2 residues E2292, F2293, and P2334 did not result in any

**Table 2 Data collection and refinement statistics.**

|  | ARM-F15X-native | ARM-F15X-Se-Met |
|---|---|---|
| Data collection |  |  |
| Space group | P1 | P2₁ |
| Cell dimensions |  |  |
| $a, b, c$ (Å) | 52.45, 70.77, 75.87 | 52.89, 135.83, 75.79 |
| $\alpha, \beta, \gamma$ (°) | 96.47, 109.44, 103.99 | 90, 110.38, 90 |
| Resolution (Å) | 48.4-2.7 (2.77-2.7)[a] | 49.6-2.6 (2.67-2.6)[a] |
| $R_{pim}$ | 0.053 (1.305) | 0.062 (1.395) |
| $R_{merge}$ | 0.074 (1.837) | 0.158 (3.589) |
| $I / \sigma I$ | 7.3 (0.6) | 8.0 (0.5) |
| $I / \sigma I$[b] | 8.8 (1.3) | 9.4 (0.9) |
| $CC_{1/2}$ | 0.997 (0.300) | 0.998 (0.159) |
| $CC_{1/2}$[b] | 0.997 (0.479) | 0.998 (0.355) |
| Completeness (%) | 96.7 (93.5) | 100 (99.8) |
| Completeness (%)[b] | 80.0 (38.1) | 83.7 (28.4) |
| Redundancy | 2.8 (2.8) | 7.4 (7.5) |
| B Wilson |  | 68.99 |
| Refinement |  |  |
| Resolution (Å) |  | 49.6-2.6 |
| No. reflections |  | 25870 |
| $R_{work}/R_{free}$ |  | 0.189/0.259 |
| No. atoms |  | 7383 |
| Protein |  | 7241 |
| Ligand/ion |  | 2 |
| Water |  | 140 |
| Protein residues |  | 945 |
| B-factors |  | 87.2 |
| Protein |  | 87.7 |
| Ligand/ion |  | 64 |
| Water |  | 62.5 |
| R.m.s. deviations |  |  |
| Bond lengths (Å) |  | 0.013 |
| Bond angles (°) |  | 1.58 |
| PDB entry code |  | 7BDX |

The anisotropic diffraction data for ARM-F15X-native were truncated using STARANISO to include all valid data (reflections with I/σ(I) of 1.2) to resolutions of: 2.68, 3.22 and 2.50 Å along the 0.705a* −0.402b* −0.584c*, 0.180a* +0.970b* −0.164c*, 0.196a* −0.262b* +0.945c* directions, respectively. Similarly, the anisotropic diffraction data for ARM-F15X-SeMet were truncated using STARANISO to include all valid data to resolutions of: 2.69, 2.91, and 2.47 Å along the 0.861a* −0.508c*, b*, 0.052a* +0.999c* directions, respectively.
[a]Values in parentheses are for highest-resolution shell.
[b]Values after truncation by STARANISO.

loss of binding, despite F2293 and P2334 being buried at the interface with HSF2BP. These residues were mutated into leucine: their hydrophobic character was conserved, which might explain the lack of associated binding defect (Fig. 4d). In summary, the impact of human polymorphisms at the HSF2BP-BRCA2 interface was characterized, and a set of mutations of highly conserved residues was identified that severely decreased the interaction between these two proteins in cells.

**Functional interaction with HSF2BP requires Brca2 exon 12.** Previously, we showed that excising exon 12 from *BRCA2* in human cells, mimicking a naturally occurring *BRCA2* splice form, renders them completely resistant to the inhibitory effect HSF2BP has on HR in the context of DNA interstrand crosslink repair[15]. This suggested complete disruption of the functional interaction between HSF2BP and BRCA2Δ12. Analysis of our crystal structure revealed that exon 12 encodes motif 1, whereas exon 13 encodes motif 2 (Fig. 5a). Thus, deleting exon 12 should lead to the loss of more than half of the binding interface, and also to the absence of resulting tetramerization of the ARM domain. To test this hypothesis, we measured the affinity of HSF2BP for the F15X

peptide which has the sequence encoded by exon 12 deleted, named F15XΔ12 (D2312-E2342). This affinity is in the micromolar range, that is 1000-fold weaker than the affinity of HSF2BP for F0 (Fig. 5b). Moreover, the stoichiometry of the interaction is now 1, demonstrating that each HSF2BP molecule binds to its own BRCA2 peptide. Further analysis of the ARM-F15XΔ12 complex using SEC consistently showed that the ARM domain does not oligomerize upon binding to F15XΔ12 (Fig. 5c). To test the effect of exon 12 loss under physiological conditions, we created the *Brca2* exon 12 deletion in mouse embryonic stem (ES) cells, where HSF2BP is expressed natively[16]. To further validate the specificity of our system, we also engineered another *Brca2* in-frame exon excision, deleting exons 12–14 which encode all of BRCA2 residues involved in the interaction with HSF2BP (Fig. 5d). In addition to the different *Brca2* exon excisions we homozygously knocked-in *GFP* expression sequence at the 3′ end of *Hsf2bp* or *Brca2* coding sequence[16,37]. This allowed us to study HSF2BP-BRCA2 interactions under native expression levels, and at the same time take advantage of the highly efficient GFP nanobody precipitation, thus reducing the chance of missing any possible residual interaction between the proteins, while minimizing non-specific background associated with indirect immunoprecipitation. To avoid non-linear amplification in immunoblotting, we used fluorescently labeled secondary antibodies instead of enzymatic detection. Pull-downs from cells producing full-length, Δ12 or Δ12–14 BRCA2-GFP from engineered homozygous alleles revealed near-complete (by 95 ± 3%, $n = 4$) and complete abrogation of HSF2BP co-precipitation in Δ12 and Δ12–14, respectively (Fig. 5d). Pull down of HSF2BP-GFP from *Hsf2bp*^GFP/GFP^*Brca2*^wt/wt OR Δ12/Δ12 OR Δ12-14/Δ12-14^ ES cells revealed an essentially complete disruption (97–99%, $n = 2$) of co-precipitation in BRCA2-Δ12, and only background signal for BRCA2-Δ12–14 (Fig. 5e). Consistently, pull down of GFP-HSF2BP from HEK293T or HeLa cells showed no binding to several FLAG-F9 variants (Supplementary Fig. 1d). We also tested co-precipitation between human proteins in HeLa cells overproducing GFP-HSF2BP and producing full-length, Δ12 or Δ12–14 BRCA2 from engineered native alleles. Human BRCA2 and HSF2BP behaved similar to the mouse proteins (Fig. 5f) and consistent with the functional experiments we described before[15]. Thus, in agreement with our biophysical data, loss of *Brca2* exon 12 strongly decreased interaction with HSF2BP. To evaluate its effect on HSF2BP-BRCA2 in functional contexts in cells, we analyzed HSF2BP-GFP diffusion in living cells and its recruitment to ionizing radiation-induced nuclear foci, using the same engineered *Hsf2bp* and *Brca2* allele combinations in mES cells. Characteristic (BRCA2-like) constrained diffusion of HSF2BP-GFP we described before[16] was dramatically affected by exons 12–14 deletions; in particular the slow-diffusing and immobile species were gone (Supplementary Movies 1–3). We further noted that in *Brca2* Δ12 and Δ12–14 cells, HSF2BP-GFP fluorescence intensity in the nucleus was reduced, and more fluorescence was observed in the cytoplasm, which made it altogether impossible to apply the quantitative single particle tracking analysis we used before. We observed a similar reduction in nuclear fluorescence and co-localization of HSF2BP-GFP with RAD51 in ionizing radiation-induced nuclear foci in immunofluorescence experiments (Fig. 5g, h). Taken together, this and the functional experiments in human *BRCA2*^Δ12/Δ12^ cells, indicate that exclusion of the BRCA2 domain encoded by exon 12 leads to a severe defect in the interaction with HSF2BP in cells.

**High-affinity HSF2BP-BRCA2 binding dispensable for meiotic HR.** Our biophysical data and functional experiments validated Brca2-Δ12 as a model to determine the functional importance of the

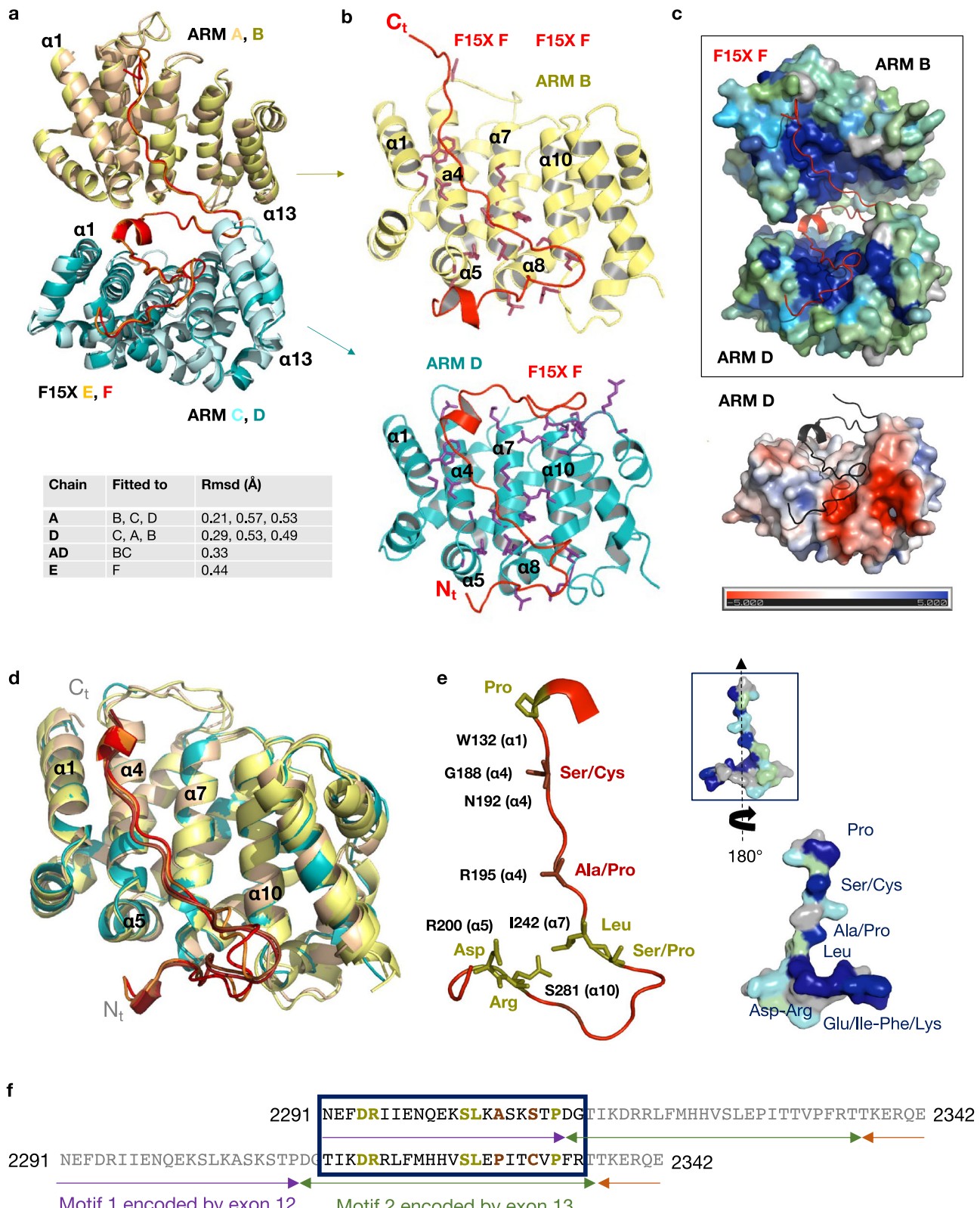

high-affinity interaction between BRCA2 and HSF2BP and of the resulting HSF2BP oligomerization. To study this in the context of meiotic HR, we created a *Brca2^{Δ12/Δ12}* mouse model (Fig. 6a–c). Consistent with the robust proliferation of the engineered BRCA2 Δ12 HeLa and mES cells, *Brca2^{Δ12/Δ12}* mice were viable, born at Mendelian ratios, and did not show any overt phenotypes. Contrary to our expectation, the Δ12 mutation did not phenocopy the *Hsf2bp*

knockout, as not only females, but also males were fertile, with normal sperm counts and a significant increase rather than a reduction in testis weight (Fig. 6d–g). Morphology of testis tubule sections was normal (Supplementary Fig. 8a). Molecular analysis of the meiotic prophase progression did not reveal any major defects: the small (6–18%) reductions in the number of recombinase foci were only significant for RAD51 in leptotene and zygotene

**Fig. 3 The same conserved ARM surface interacts with both the N-terminal and C-terminal regions of the BRCA2 peptide, through a 23 aa motif.**
**a** Superimposition of the complexes formed by two ARM domains and a BRCA2 peptide. The Cα root-mean-square-deviation (Rmsd) values calculated between the different chains are recapitulated in the lower part of the panel. Additional views of the superimposed individual chains are displayed in Supplementary Fig. 7. **b** Zoom on the interfaces between ARM chain B and the C-terminal region of BRCA2 F15X chain F (upper view) and ARM chain D and the N-terminal region of BRCA2 F15X chain F (lower view). ARM residues that are either involved in hydrogen bonds or salt bridges with BRCA2, or buried (by more than 30 Å$^2$) at the interface with BRCA2, are represented by colored sticks. They are labeled in Supplementary Fig. 7b, c. **c** Surfaces of ARM domains, colored as a function of sequence conservation (upper panel) or electrostatic potential (lower panel). In the upper panel, surfaces of ARM chains B and D are colored as a function of conservation scores calculated by Consurf[65]. High, medium, weak, and no conservation are indicated in dark blue, cyan, green, and gray, respectively. BRCA2 F15X F is represented as a red ribbon. In the lower panel, the surface of ARM chain D is colored from red (negatively charged) to blue (positively charged) and the N-terminal region of BRCA2 F15X F is displayed as a black ribbon. **d** Superimposition of the four ARM domains and their BRCA2-interacting peptides. The four ARM structures were aligned, and their BRCA2-interacting fragments are displayed. **e** Representation of the 3D structure of the BRCA2 motif binding to ARM domains. Residues strictly conserved between motif 1 and motif 2 are displayed as olive sticks. They correspond to D2294/2317, R2295/2318, S2303/2326, L2304/2327, and P2311/2334. Residues that are similar between the two motifs and conserved in BRCA2 are displayed as brown sticks. They correspond to A2306/P2329 and S2309/C2332 (mutated in T2332 in our construct to avoid oxidation of this residue). A set of conserved residues from ARM interacting with BRCA2, as defined in **b**, are indicated in black next to the BRCA2 residues when they directly interact with these residues. A boxed surface view of the peptide, in the same orientation as the cartoon view, is colored by conservation as in **c**. Turning this surface view by 180° reveals the conservation of the surface binding to ARM, including the conserved residues of motifs 1 and 2.
**f** Sequence alignment of motif 1, interacting with ARM chains C and D, and motif 2, interacting with ARM chains A and B. Conserved residues are colored as in **e**. Motif 1 is encoded by exon 12, whereas motif 2 is encoded by exon 13.

(Fig. 6i–h). In addition, the frequency of MLH1 foci indicating meiotic crossover sites after successful HR was normal (Fig. 6k, Supplementary Fig. 8b). These results were not consistent with the proposed role of HSF2BP as a BRCA2 localizer in meiosis. We hypothesized that given the high-affinity interaction revealed by our ITC experiments, the dependence may be opposite: BRCA2 may bring HSF2BP to the double-strand breaks (DSBs). However, the number of HSF2BP foci was not significantly affected by *Brca2* exon 12 deletion, and for BRME1 foci only small (but statistically significant) reductions in leptotene and pachytene were measured (reduced by 23% and 6%, respectively Fig. 6l–n). Consistent with the robust accumulation of both HSF2BP and RAD51 to the meiotic DSB chromatin in *Brca2*$^{\Delta12/\Delta12}$ testis, both proteins co-precipitated with BRCA2, and the effect of disrupting HSF2BP-BRCA2 interaction as revealed by our structure and variant mapping was smaller in the testis co-IPs (Fig. 6c). The difference can be caused by the potential contributions of other interactions, which are absent or substantially sub-stoichiometric in the experiments with purified or highly overproduced proteins (Fig. 5, Supplementary Fig. 1). Together with the weakened binding via motif 2 such compensatory interactions (e.g., mediated by other binding partners, chromatin, or post-translational modifications) may mask the meiotic defect resulting from *Brca2* exon 12 loss in the genetic background of the inbred laboratory mouse strain we used.

## Discussion
In this paper, we analyzed the structural properties and functional consequences of the BRCA2-HSF2BP interaction and tested the emerging model of its involvement in meiosis. The essential roles of BRCA2 and HSF2BP in meiotic HR have been clearly demonstrated previously. BRCA2 interacts with both RAD51 and DMC1 recombinases, is required for their accumulation at meiotic DSB in mice and stimulates their activity in vitro. But how BRCA2 balances its activity with respect to RAD51 and DMC1 and integrates with the meiotic-specific HR machinery in a timely manner during meiosis remains unclear. Direct data on its behavior in meiocytes is scarce, and mechanistic models are mostly based on extrapolation. The proposed role of the recently identified HSF2BP, required for RAD51 and DMC1 accumulation at meiotic DSBs during spermatogenesis in mice[16], is to bring BRCA2 to meiotic DSBs[27]. We tested this hypothesis by disrupting the HSF2BP-binding region of BRCA2 in mice.

We first characterized the BRCA2-HSF2BP interaction in vitro. We had previously identified that the region of BRCA2 binding to HSF2BP is located between its BRC repeats and its C-terminal DNA binding domain (Fig. 1b and ref. [16]). Our structural analysis revealed that this BRCA2 region contains a duplicated motif, which was not previously recognized from its primary amino acid sequence. Each motif binds to the same residues of an Armadillo domain of HSF2BP (Fig. 3e). By itself, the Armadillo domain dimerizes through a conserved surface formed by helices α1 to α3 (Fig. 2d) and presents on each monomer a large conserved groove, which indicates a binding site for functionally important partners (Fig. 3c). Many Armadillo domains interact through their concave surface with largely disordered partners[38]. The Armadillo domain of HSF2BP contains four Armadillo repeats (Supplementary Fig. 7a). Altogether, they form a positively charged groove delimited by helices α1, α4, α7, α10, and α13. This groove is able to recognize a 23 aa motif located in a conserved and disordered region of BRCA2. Because this motif is duplicated in BRCA2, and each motif binds to a different ARM dimer, the interaction triggers further oligomerization of the Armadillo domain into a tetramer. The affinity of BRCA2 for HSF2BP is 1 nM, which is significantly higher than the affinities yet measured between BRCA2 disordered regions and its partners PALB2, RAD51, and PLK1, all between 100 and 1000 nM[33–35]. However, after deleting motif 1 encoded by exon 12, motif 2 alone binds to HSF2BP with a micromolar affinity and is unable by itself to trigger oligomerization of the ARM domain into a tetramer (Fig. 5b).

Consistent with our in vitro study, we observed in mouse ES cells that deletion of exon 12, coding for motif 1, causes a severe decrease in the BRCA2-HSF2BP interaction, and deletion of exons 12 to 14, coding for motifs 1 and 2, completely abolishes this interaction (Fig. 5d, e). We previously showed that HSF2BP mutation R200T abolishes localization of the GFP-tagged HSF2BP protein to mitomycin C-induced repair foci[16]. Consistently, we now demonstrated that the BRCA2 region encoded by exon 12 is responsible for HSF2BP localization at irradiation-induced DSBs (Fig. 5g). We further previously reported that, in human cells, excising exon 12 from *BRCA2*, mimicking a naturally occurring *BRCA2* splice form, rendered them completely resistant to the inhibitory effect HSF2BP has on HR in the context of DNA interstrand crosslink repair[15].

Based on these structural, biochemical, and functional experiments, we developed a *Brca2*$^{\Delta12}$ mouse line to test the emerging model, which posits that HSF2BP-BRME1 complex acts as a meiotic localizer of BRCA2[25,27], and thus predicts that

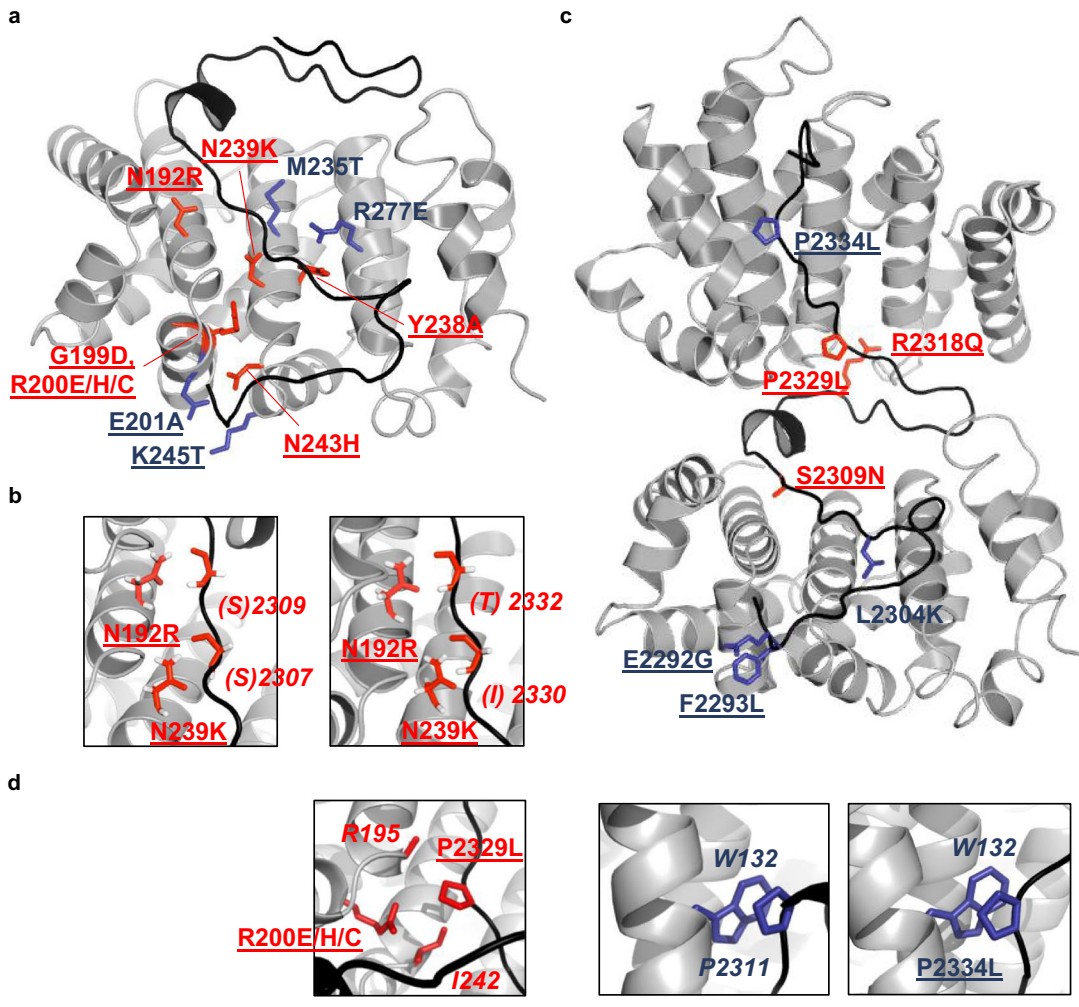

**Fig. 4 Crystal structure explains the effect of human SNP variants on HSF2BP-BRCA2 interaction. a** 3D view of the ARM chain D in complex with motif 1 of BRCA2 F15X chain F. The ARM and BRCA2 protein fragments are displayed as gray and black ribbons, respectively. The side chains substituted for coimmunoprecipitation studies are displayed in sticks, and colored as in Fig. 1a (red: no binding; blue: residual binding with the residue name underlined when wild-type binding was observed). **b** Zoom views on residues forming a hydrogen bond network between ARM and BRCA2 F15X. Asparagine side chains from ARM interact with backbone atoms from BRCA2 F15X. The side chains of BRCA2 are not displayed; consistently, the BRCA2 residue names are in brackets. Hydrogens were added to the crystal structure for clarity. **c** 3D view of the ARM chains B and D in complex with the BRCA2 F15X chain F. The ARM and BRCA2 protein fragments, as well as the mutated side chains, are displayed as in **a**, except that full labels are shown for BRCA2 residues. **d** Zoom views on BRCA2 F15X mutations P to L and their interacting residues in ARM chains B and D. Residues marked in italics have not been mutated in our study.

disengaging BRCA2 from HSF2BP will phenocopy *Hsf2bp* deficiency. However, we could not detect any major differences in meiosis in *Brca2^Δ12/Δ12^* mice compared to *Brca2^+/+^* (Fig. 6). Not only females, but also males were fertile, had a normal sperm count and increased testis weights. While the latter is opposite to the greatly reduced testis sizes in *Hsf2bp* knockout and hard to explain, we noted a progressive increase in testis, epididymis, and body weights from +/+ to Δ/+ to Δ/Δ genotypes, although the differences in body and epididymis weights are not as notable or not even statistically significant, as was observed for testes weights. Regarding our detailed immunocytochemical analyses of the progression of homologous chromosome pairing and meiotic DSB repair during meiotic prophase, it is clear that these events occurred grossly normal, except for a significant although small reduction in the number of RAD51 foci in early stages. Still, the number of DMC1 foci, as well as MLH1 foci were normal indicating normal meiotic crossover formation after successful HR. Despite the clear reduction in BRCA2-HSF2BP interaction in *Brca2^Δ12/Δ12^* testes, HSF2BP foci numbers were not significantly

reduced. A small decrease in the number of BRME1 foci in *Brca2^Δ12/Δ12^* leptotene spermatocytes may be indicative of some interdependence between BRCA2, HSF2BP, and BRME1, but hard to reconcile with the normal HSF2BP foci within the proposed models. Altogether, these analyses show that the high-affinity interaction between BRCA2 and HSF2BP, together with the oligomerization of HSF2BP triggered by this interaction, are not essential for HR in meiosis.

The BRCA2 localizer function of HSF2BP was suggested primarily based on the observation of the localization of recombinant GFP-tagged BRCA2 fragments produced by electroporation of expression constructs into wild-type and HSF2BP-deficient testis[27]. In these experiments, a BRCA2 fragment including the HSF2BP-binding region and the C-terminal ssDNA binding domain co-localized with RPA2 at DSB sites in an HSF2BP-dependent manner[27]. The presence of various ssDNA-binding proteins (SPATA22, MEIOB, and RPA) in HSF2BP and BRME1 immunoprecipitates further supported the model and led to the suggestion that HSF2BP and BRME1 act as adaptors, anchoring

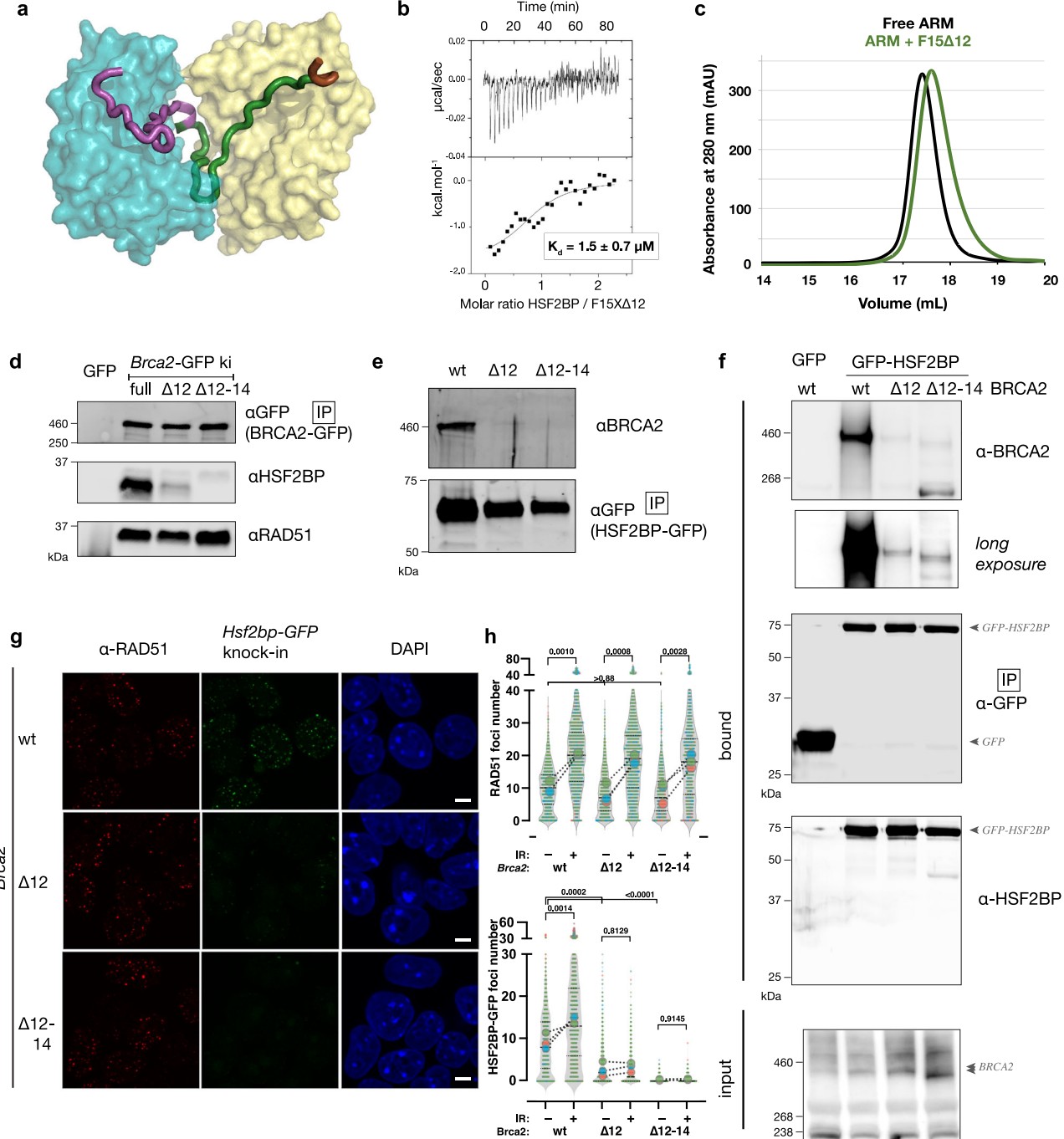

**Fig. 5 Deletion of *Brca2* exon 12 disrupts HSF2BP-BRCA2 interaction in cells. a** Structure of the HSF2BP dimer interacting with the BRCA2 peptide colored by encoding exon number. **b** ITC experiment with HSF2BP and a truncated variant of F15X peptide missing residues encoded by exon 12 (F15XΔ12). **c** ARM and the complex between ARM and F15XΔ12 analyzed by analytical gel filtration (column: Superose 6 Increase 10/300 GL). **d** Immunoblot analysis of proteins co-precipitated with anti-GFP nanobody beads from mES cells containing homozygous *Brca2-GFP* allele without (full) or with deletions of exon 12 or exons 12–14. The experiment was performed four times with similar results. **e** Immunoblot analysis of proteins co-precipitated with anti-GFP nanobody beads from double knock-in mES cells containing homozygous *Hsf2bp-GFP* and *Brca2* alleles without or with deletion of exons 12 or 12–14. The experiment was performed two times with similar results. **f** Immunoblot analysis of proteins co-precipitated with anti-GFP nanobody beads from HeLa cells stably overproducing GFP-HSF2BP or GFP control, and in which *BRCA2* allele was modified by excision of exons 12 or exons 12–14, or unmodified wild-type (wt) *BRCA2*. **g** *Hsf2bp*^GFP/GFP^*Brca2*^wt/wt^, *Hsf2bp*^GFP/GFP^*Brca2*^Δ12/Δ12^, and *Hsf2bp*^GFP/GFP^*Brca2*^Δ12-14/Δ12-14^ mES cells were irradiated with 8 Gy, fixed after 2 h recovery, immunostained with anti-RAD51 antibody, mounted with DAPI, and imaged using laser confocal microscope. HSF2BP-GFP was detected by direct fluorescence. Maximum projection of three confocal slices (0.5 μm apart) is shown. Scale bar = 5 μm. **h** Quantification of HSF2BP-GFP and RAD51 foci from the experiments as the one shown in **g**. Data from three independent experiments are plotted following the SuperPlots approach[66]: symbol colors indicate biological replicas (*n* = 3), small symbols show number of foci per nucleus, violin plot shows the combined frequency distribution, with lines indicating median and quartiles; large circles indicate means within replicas, dotted lines visualize changes after irradiation. Replica means were compared by one-way ANOVA with Tukey multiple comparison test, *p*-values are indicated.

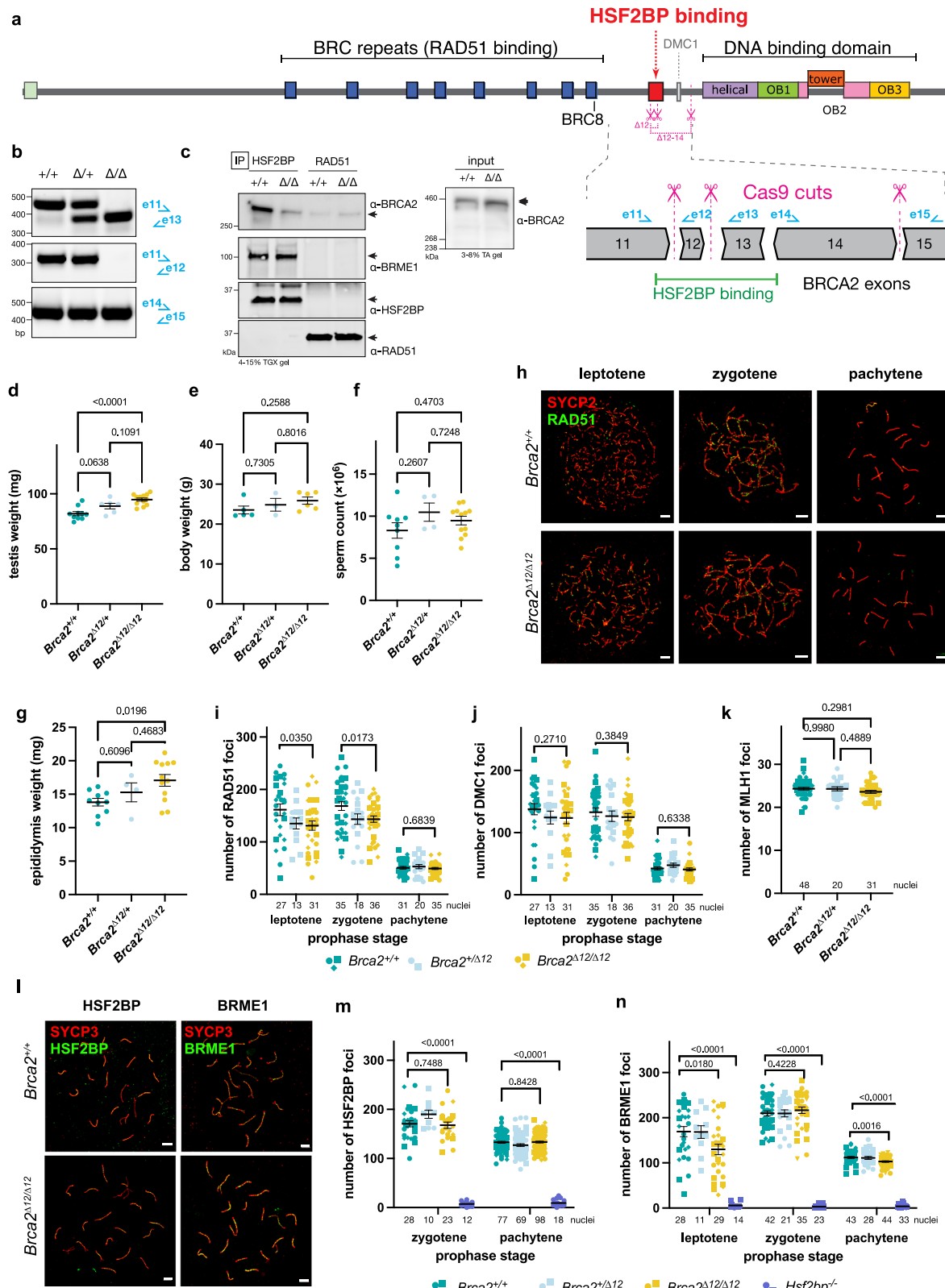

BRCA2 to the protected ssDNA. Such adaptors have not been identified in somatic HR. Both BRCA2 and its equally important partner PALB2, also involved in meiotic HR[39], are well equipped for the recruitment to resected DSBs with a set of DNA-, RAD51- and chromatin-binding domains, and can stimulate RAD51 and/ or DMC1 activity autonomously in vitro. As a localizer, HSF2BP neither mimics DNA as does the BRCA2-interacting protein

DSS1[40], nor interacts with ssDNA or dsDNA (Supplementary Fig. 9; see also ref. [22]). However, even with these reservations regarding the localizer model, we fully expected the HSF2BP-binding domain of BRCA2 to be essential for meiotic HR, because it is conserved, and no somatic function could be assigned to it by several previous studies in human cancer cells[41,42]. The high evolutionary conservation of the BRCA2-HSF2BP interaction,

**Fig. 6 Meiotic phenotype of *Brca2* exon 12 deletion mouse model. a** Schematic depiction of the domain composition of the BRCA2 protein and the exons 11-15 encoding HSF2BP-binding and DMC1-binding domains. Introns are not drawn to scale; different exon phases are indicated by the shape of the boundary. Location of Cas9 cut sites for exon 12 and exons 12–14 excision is shown. **b** RT-PCR on cDNA from mouse testis with indicated genotypes confirming the loss of exon 12; primer locations are shown on the exon scheme in **a**. The PCR was performed twice with the same results. **c** Immunoblot analysis of proteins precipitated from *Brca2*[+/+] and *Brca2*[Δ12/Δ12] mouse testes using anti-HSF2BP antibody and, as control, with anti-RAD51 antibodies; and the input samples; performed as described in "Methods" section. The experiment was performed four times with similar results. **d** testis weight, **e** bodyweight, **f** sperm count, and **g** epididymis weight in *Brca2*[Δ12/Δ12] and control mice. $n = 5$ animals for *Brca2*[+/+], $n = 6$ for *Brca2*[Δ12/Δ12], $n = 3$ (testis weight and bodyweight), and $n = 2$ (epididymis weight and sperm count) for *Brca2*[Δ12/+]. Mean, s.e.m, and $p$ values from one-way ANOVA with Tukey test are indicated. **h–n** Immunofluorescent analysis of meiotic protein localization on spermatocyte spreads from *Brca2*[Δ12/Δ12] and control mice. Representative images (**h, l**; scale bars = 5 μm) and quantification of RAD51 (**i**), DMC1 (**j**), MLH1 (**k**), HSF2BP (**m**), and BRME1 (**n**) foci are shown with mean and s.e.m. indicated with error bars. Symbol shapes designate individual animals: $n = 3$ animals for *Brca2*[+/+] and *Brca2*[Δ12/Δ12], $n = 2$ for *Brca2*[Δ12/+], $n = 2$ (BRME1), and $n = 1$ (HSF2BP) for *Hsf2bp*[−/−]. Mean, s.e.m, and $p$ values from two-tailed unpaired $t$-test for selected pairwise comparisons within prophase stages are indicated. The number of analyzed nuclei is indicated for each genotype and stage.

initially evident from sequence analysis and interchangeability of human and frog HSF2BP proteins in biochemical experiments[15], is further emphasized by the structure we solved. The BRCA2-binding surface is a particularly highly conserved part of HSF2BP (Fig. 3c), and the cryptic BRCA2 repeats, to which it binds with a remarkable structural symmetry, are also highly conserved (Supplementary Fig. 2). This means that the oligomerization-inducing interaction between the two proteins is under strong selective pressure and thus functionally relevant. Further investigation of the role of HSF2BP and the HSF2BP-BRCA2 interaction in meiosis, and outside of it, is now required.

Our crystal structure also revealed the mechanism of BRCA2-induced HSF2BP oligomerization observed in the biochemical experiments (Fig. 2, refs. [16,25]). The main contribution to the nanomolar affinity we measured comes from the large 2740 Å$^2$ interface between one BRCA2 peptide and two HSF2BP ARM domains from two different ARM dimers. A second contribution is made by additional contacts between two ARM dimers (Supplementary Fig. 5b). As the corresponding interface is small (350 Å$^2$ vs. twice 2740 Å$^2$ in one BRCA2-ARM complex), its contribution to the thermodynamic parameters we measured by ITC is not expected to be significant, and its role may be in establishing proper spatial orientation of the ARM domains. Thus, HSF2BP is able to oligomerize through several distinct mechanisms: it was previously reported that full-length HSF2BP contains an N-terminal domain forming coiled coils[25]; we now show how its armadillo domains dimerize and further tetramerize upon binding to BRCA2. Each oligomerization mechanism can be under separate regulatory control, allowing HSF2BP to serve as a versatile and potent agent increasing the local concentration and/ or modifying the oligomerization state of BRCA2. This can play a positive role in some contexts and be detrimental in others. For example, we previously found that HSF2BP, when produced ectopically in somatic cancer cells, interferes with the role of BRCA2 in DNA interstrand crosslink repair by causing its degradation[15]. The findings we report here suggest that this degradation, which is mediated by the p97 segregase and is proteasome-dependent, could result from HSF2BP-induced BRCA2 aggregation.

Altogether, we conclude that the evolutionarily conserved high-affinity oligomerization-inducing interaction mode between HSF2BP and BRCA2 we described in this paper is not required for the recruitment of RAD51 and DMC1 strand exchange proteins and for productive HR in meiosis. While this does not rule out that co-localization of HSF2BP and BRCA2 at meiotic DSBs via alternative compensatory interactions is essential, it raises the question as to why the peculiar repetitive structure evolved and remained conserved. It also opens the possibility that the meiotic function of HSF2BP associated with fertility may result from other interactions, mediated either by its N-terminal coiled-coil

domain, which binds BRME1, by the conserved surface of its ARM domain binding to other proteins with sequences similar to the cryptic repeat our structure revealed in BRCA2, or by other parts of the ARM domain.

## Methods

**Cells, DNA constructs, and transfection.** HeLa (human cervical adenocarcinoma, female origin) and HEK293T (human embryonic kidney, female origin) cells were cultured in DMEM supplemented with 10% FCS, 200 U/ml penicillin, 200 μg/ml streptomycin. mES cell lines were derived from the IB10 cell line, which is a subclone of E14 129/Ola from male origin[43], specific pathogen free. Cells were cultured on gelatinized plastic dishes (0.1% gelatin in water) as described before[44] at atmospheric oxygen concentration in media comprising 1:1 mixture of DMEM (Lonza BioWhittaker Cat. BE12-604F/U1, with Ultraglutamine 1, 4.5 g/l glucose) and BRL-conditioned DMEM, supplemented with 1000 U/ml leukemia inhibitory factor, 10% FCS, 1x NEAA, 200 U/ml penicillin, 200 μg/ml streptomycin, and 89 μM β-mercaptoethanol.

Expression constructs for producing point mutation and truncation variants of HS2BP and BRCA2 in human cells were engineered as described before[16] in the PiggyBac vectors by Gibson assembly. For transient expression in HEK293T cells plasmid DNA was transfected using calcium precipitation method or PEI transfection. For stable integration into HeLa cells, PiggyBac expression vectors were co-lipofected together with PiggyBac transposase plasmid (hyPBase[45]) with Lipofectamine 3000 (Thermo Fisher).

GFP knock-in alleles in mES cells were engineered using the previously described CRISPR/Cas9-stimulated approach. All gRNAs were cloned into a derivative of pX459 vector. Excision of BRCA2 exon 12 from mES cells was performed with gRNAs targeting the same sequences in intron 11 and intron 12 as those used to produce *Brca2*Δ12 mouse alleles (see below). Excision of exons 12–14 was performed with gRNAs targeting the same sequence in intron 11, and the sequence in intron 14 CCAACCAGCCCGGTCAAGTT. IB10 or the previously described[37] *Brca2*[GFP/GFP] were used as parental cell lines. Excision of exons 12–14 from HeLa cells was performed with the same gRNA target in intron 11 as used before and the following target in intron 14: AGGAGAGCATGTAAACTTCG. Cell lines and other biological materials generated for the study can be shared upon reasonable request, subject to institutional MTA. Cells were genotyped by PCR. Excision was further confirmed by RT-PCR analysis of the first-strand cDNA produced from total mRNA with oligo-dT primers with SuperScript II polymerase (Invitrogen).

**Immunoprecipitation and immunoblotting.** Cells were washed twice in ice-cold PBS and lysed in situ in NETT buffer (100 mM NaCl, 50 mM Tris pH 7.5, 5 mM EDTA pH 8.0, 0.5% Triton-X100) supplemented immediately before use with protease inhibitor cocktail (Roche) and 0.4 mg/ml Pefabloc (Roche) (NETT++); 450 μl NETT++ buffer was used per 145 mm dish ES cells, 1 ml for HeLa and HEK293T. After 5–10 min, cells were scraped off and collected in 1.5 ml micro-centrifuge tubes; lysis was continued for additional 20–30 min on ice, then mixtures were centrifuged (15 min, 4 °C, 14,000 × $g$) and the supernatant (input) was added to washed anti-GFP beads (Chromotek). Beads and lysates were incubated 2–4 h at 4 °C while rotating, washed three times in NETT++ buffer and bound proteins were eluted by boiling in 2x Sample buffer. Immunoblotting was performed following standard procedures with the following antibodies: anti-GFP mAb (Roche, #11814460001), anti-GFP pAb (Abcam #ab290 and Invitrogen #A11122), anti-RAD51[46], anti-BRCA2 mAb Ab1 OP-95 (Millipore #OP95), anti-BRCA2 (Abcam #27976), anti-Flag (M2 antibody, Sigma, F3165 and F1804). For quantitative immunoblotting fluorescently labeled secondary antibodies were used: anti-mouse CF680 (Sigma #SAB460199), anti-rabbit CF770 (Sigma #SAB460215); membranes were scanned using Odyssey CLx imaging system (LI-COR).

For testis immunoprecipitation, one whole testis was homogenized in 2 ml of NETT++ or RIPA++ (50 mM Tris HCl pH 7.5, 150 mM NaCl, 1% Triton-X100, 0.5% Na-deoxycholate, 0.1% SDS, 1 mM EDTA, 1 mM EGTA, supplemented immediately before use with cOmplete protease inhibitor cocktail (Roche, 11836145001) and 0.4 mg/ml Pefabloc (Roche)) buffer by 3–5 s pulse in Polytron homogenizer. The lysate was allowed to settle on ice for 20–40 min, transferred to mini-centrifuge tubes and cleared by 15 min centrifugation at maximum speed (~13,000 × g) at 4 °C. A 70 μl aliquot of the cleared supernatant was mixed with 70 μl of the 2× sample buffer and denatured for 5 min at 95 °C (input sample). The remainder of the supernatant was divided into two fractions, which were incubated at 4 °C for 4 h with homemade affinity-purified anti-HSF2BP (SY8127, 40 μg/IP), anti-RAD51 (2037, 2 μl/IP) or pre-immune sera (4 μl/IP) cross-linked using DMP to 50 μl magnetic protein A beads (BioRad SureBeads #161-4013). After incubation, beads were washed three times for 5 min at 4 °C with 1 ml lysis buffer, and then incubated for 5 min at 95 °C with 35 μl of 2× sample buffer to elute the bound proteins. All of the eluate was run in a single lane of a 4–15% TGX SDS-PAGE gel (BioRad #456-1084) and transferred to PVDF membrane, which was cut into three fragments: the top part was immunoblotted with sheep anti-BRCA2 antibody[47] followed by ECL detection with HRP-conjugated secondary antibody (ThermoFisher, A16041), the middle and bottom parts with anti-BRME1 and a mixture of anti-RAD51 and anti-HSF2BP followed by fluorescent detection. Input samples were analyzed separately on a low-percentage tris-acetate SDS-PAGE gel (3–8% NuPAGE, Thermo Fisher EA0375), suitable for detection of high-molecular weight proteins, such as BRCA2 in total cell lysates.

**Protein expression and purification.** Human full-length HSF2BP WT and R200T were expressed using a pETM11 (6xHis-TEV) expression vector in *E. coli* BL21 DE3 Rosetta2 cells, and purified as previously reported (Brandsma et al., *Cell Rep* 2019). The armadillo domain of HSF2BP, from aa 122 to aa 334, which we will further name ARM, was similarly expressed using a pETM11 (6xHis-TEV) expression vector in BL21 DE3 Rosetta2 cells and purified as full-length HSF2BP.

Human BRCA2 fragment F0 was expressed using a pET-22b expression vector as a fusion protein comprising BRCA2 from aa 2213 to aa 2342 (including mutation C2332T to avoid oxidation problems), a TEV site, a GB1 and 6xHis tag, in *E. coli* BL21 DE3 Star cells. The BRCA2 gene was optimized for expression in bacteria and synthesized by Genscript. In addition, smaller BRCA2 fragments were produced using the same strategy: $F_{NMR}$ and F15X corresponding to aa 2252 to aa 2342 and aa 2291 to aa 2343, respectively. The smallest fragment F15XΔ12, from aa 2312 to aa 2342, was synthesized by Genecust.

For NMR analysis, $^{15}$N-labeled or $^{15}$N- and $^{13}$C-labeled BRCA2 fragments were produced in *E. coli* BL21 DE3 Star cells grown in M9 medium containing either 0.5 g/l $^{15}$NH$_4$Cl or 0.5 g/l $^{15}$NH$_4$Cl and 2 g/l $^{13}$C-glucose, respectively. For crystallography, selenomethionine(SeMet)-labeled ARM and F15X were produced in transformed *E. coli* BL21(DE3) Rosetta2 (ARM) and Star (F15X) cells, respectively, grown in minimum medium (16 g of Na$_2$HPO$_4$, 4 g of KH$_2$PO$_4$, 1 g of NaCl, 0.5 g of EDTA, 0.4 g of FeCl$_3$, 0.04 g of ZnCl$_2$, 0.006 g of CuCl$_2$ 6 H$_2$O, 0.005 g of CoCl$_2$, 0.005 g of H$_3$BO$_3$, 4 g of glucose, 20 mg of thiamine, 20 mg of biotin, 1 g of (NH$_4$)$_2$SO$_4$, 0.5 g of MgSO$_4$, and 0.1 g of CaCl$_2$ in 1 l of MilliQ), supplemented with 200 mg of each amino acid and 125 mg of SeMet per 1 l of medium.

Protein expression was induced with 0.2 mM IPTG at an OD$_{600}$ of 0.6, and incubated further ON at 20 °C (HSF2BP), or induced with 1 mM IPTG and incubated for 3 h at 37 °C (BRCA2). Harvested cells were resuspended in 25 mM Tris-HCl pH 7.5, 500 mM NaCl (HSF2BP) or 150 mM NaCl (BRCA2), 5 mM β-mercaptoethanol, EDTA-free Protease Inhibitor Cocktail (Roche) and disrupted by sonication. Lysates were supplemented with 1.5 mM MgCl$_2$ and treated by Benzonase nuclease at 4 °C for 30 min, and then centrifuged at 48,384 × g at 4 °C for 30 min. After filtration (0.4 μm), the supernatant was loaded on a chromatography HisTrap HP 5 mL column (GE Healthcare) equilibrated with the buffer Tris-HCl 25 mM. pH 8, 500 mM NaCl (HSF2BP), or 150 mM NaCl (BRCA2) and 5 mM β-mercaptoethanol. The proteins were eluted with a linear gradient of imidazole. The tag was cleaved by the TEV protease (at a ratio of 2% w/w) during an ON dialysis at 4 °C against 25 mM Tris-HCl pH 7.5, 150 mM NaCl, 5 mM β-mercaptoethanol. The protein solution was loaded on a HisTrap column and the tag-free proteins were collected in the flow through. Finally, a size exclusion chromatography was performed on HiLoad Superdex 10/300 200 pg (HSF2BP) or 75 pg (BRCA2) equilibrated in 25 mM Tris-HCl pH 7.5, 250 mM NaCl (HSF2BP) or 150 mM NaCl (BRCA2), 5 mM β-mercaptoethanol. The quality of the purified protein was analyzed by SDS-PAGE and the protein concentration was determined by spectrophotometry using the absorbance at 280 nm. The protein thermal stability was evaluated using the simplified Thermofluor assay available on the High Throughput Crystallization Laboratory (HTX Lab) of the EMBL Grenoble[48].

**BRCA2 peptide structural analysis by NMR.** The $^{15}$N/$^{13}$C-labeled F0 fragment was analyzed by 3D heteronuclear NMR. in order to assign its Hn, N, Cα, Cβ, and Co chemical shifts, and identify its binding site to HSF2BP. Therefore, 3D NMR HNCA, CBCACONH, HNCACB, HNCANNH, HNCO, and HNCACO experiments were performed on a 700 MHz Bruker AVANCE NEO spectrometer equipped with a triple resonance cryogenic TCI probe at 283 K. The data were processed using Topspin 4.0 (Bruker) and analyzed using CCPNMR 2.4[49]. Sodium

trimethyl-silyl-propane-sulfonate (DSS) was used as a chemical shift reference. Experiments were performed on a 5-mm-diameter Shigemi sample tube containing the 500 μM uniformly $^{15}$N/$^{13}$C-labeled protein, in 50 mM Hepes pH 7.0, 50 mM NaCl, 1 mM EDTA, and 95:5% H$_2$O/D$_2$O. For binding studies, 2D NMR $^1$H-$^{15}$N HSQC experiments were recorded on a 3-mm-diameter sample tube containing the $^{15}$N-labeled F0 at 100 μM in the absence and presence of the ARM domain at 100 μM, on a 950 MHz Bruker Avance III spectrometer equipped with a triple resonance cryogenic TCI probe at 283 K.

**Protein–protein interactions.** ITC experiments were performed using a high-precision VP-ITC instrument (GE Healthcare) at 293 K. To characterize the interactions between HSF2BP and BRCA2 fragments (WT, variants), the proteins were dialyzed against 25 mM Tris pH 7.5, 250 mM NaCl, 5 mM β-mercaptoethanol. The HSF2BP (or ARM domain) was in the sample cell at 10–20 μM and was titrated by BRCA2 fragments at 40–100 μM, using 10 μL injections with 210 s intervals between each injection. The first 2 μL injection was ignored in the final data analysis. The integration of the peaks corresponding to each injection. the correction for the baseline and the fit were performed using the Origin 7.0 software provided by the manufacturer, to obtain the stoichiometry (N), dissociation constant ($K_d$), and enthalpy of complex formation (ΔH) for each interaction. These data are indicated in Table 1. Two replicates were performed for each experiment.

Size-exclusion chromatography (SEC) coupled to MALS was used in order to measure the molecular masses of the complexes in solution. Therefore, HSF2BP and ARM proteins were loaded in the presence and absence of the BRCA2 fragment F15X on a Superdex 200 10/300 GL (GE Healthcare) using a HPLC Shimadzu system coupled to MALS/QELS/UV/RI (Wyatt Technology). The chromatography buffer was 25 mM Tris-HCl buffer, pH 7.5, 250 mM NaCl, 5 mM β-mercaptoethanol. The proteins were injected at 0.8–1 mg/ml in 100 μl. Data were analyzed using the ASTRA software; a calibration was performed with BSA as a standard.

SEC coupled to SAXS is available on the SWING beamline at synchrotron SOLEIL, in order to obtain a distance distribution corresponding to each sample in solution. The free HSF2BP protein as well as the complex between ARM and F15X were analyzed using a Bio SEC-3 column (Agilent) equilibrated in 25 mM Tris-HCl buffer, pH 7.5, 250 mM NaCl, and 5 mM β-mercaptoethanol. The proteins were loaded at a concentration of 6 and 10 mg/ml, in order to observe an elution peak at an OD$_{280nm}$ of 1 and 1.2 AU, respectively.

SEC was also used to characterize ARM and ARM bound to F15XΔ12. The ARM domain was loaded on a Superose 6 Increase 10/300 (GE Healthcare) alone (at 2 mg/ml) or bound to F15XΔ12 (at 5 mg/ml, ARM to peptide ratio 1:1.2) in 25 mM Tris-HCl buffer, pH 7.5, 250 mM NaCl, and 5 mM β-mercaptoethanol. The OD values from the elution of ARM alone were multiplied by 2.5 to be compared to the OD values from the elution of ARM bound to F15XΔ12.

**Crystallization and structure determination.** Prior to crystallization, the complex between ARM and F15X was loaded on a size exclusion chromatography column HiLoad Superdex 200 pg 16/600 (GE Healthcare) equilibrated in a 25 mM Tris pH 7.5, 250 mM NaCl and 5 mM β-mercaptoethanol, in order to prevent the presence of aggregates. The complex was then concentrated up to 10 mg/ml. Initial crystallization experiments were carried out at the High Throughput Crystallization[50]. Crystals were prepared for X-ray diffraction experiments using the CrystalDirect[51]. Crystals were obtained using the hanging-drop vapor diffusion method at 291 K. One μl of protein and reservoir solution containing 100 mM MgCl$_2$, 100 mM MES pH 6 and 16% (w/v) PEG 3350 were mixed. Needle crystals appeared within 3 days, were grown for 1–2 weeks and were frozen in liquid nitrogen after cryoprotection using the reservoir solution supplemented with 20% glycerol.

X-ray diffraction data were collected on the beamlines PROXIMA-1 and PROXIMA-2 at the SOLEIL synchrotron (St Aubin, France) and reduced using the XDS package[52]. First phases for a triclinic crystal form (Table 2) were obtained by molecular replacement using the program MOLREP version 11.7.03 from CCP4[53] and different homologous models. One of the models obtained by the Robetta server[54] gave the best correlation in the final translation function. These phases allowed to find the selenium substructure from a SeMet SAD dataset from the same crystal form with only ARM protein-containing selenomethionines. Later a monoclinic crystal form (Table 2) was obtained from a complex with both ARM and F15X containing selenomethionines. The collected SeMet SAD dataset (wavelength of data collection: lambda = 0.97918 Å) allowed to directly calculate phases, without external model contributions, and confirmed the initial model built in the triclinic crystal form. Se sites were found using the SHELX C/D/E suite of programs. These sites were refined using PHASER version 2.8.2 in EP mode. The resulting Se SAD phases were improved by density modification using PARROT version 1.0.4 and a model automatically build using BUCCANEER version 1.6.10 confirming the sequence attribution for ARM and F15X. The resulting model underwent iterative cycles of manual reconstruction in COOT[55] and refinement in BUSTER version 2.10.3 [56] (Table 2). At the end of the refinement, 90% and 3.5% of the residues were in favored and outlier regions of the Ramachandran plot, respectively. Few residues were not visible in the electron density (L55 in chain B, V52 and Ala53 in chain C, loop 51-55, F169, and R208 in chain D, and H33 in chain F). These residues were included in the pdb file, but with

an occupancy of 0. The final pdb file and monoclinic dataset have been deposited in the Protein Data Bank (entry code 7BDX [https://doi.org/10.2210/pdb7bdx/pdb]).

**Meiotic spread nuclei preparations and immunocytochemistry.** Meiotic testicular cells were spread as previously described[57]. For immunocytochemistry, the slides were washed in PBS (3 × 10 min), blocked in 0.5% w/v BSA and 0.5% w/v milk powder in PBS followed by staining with primary antibody which was diluted in 10% w/v BSA in PBS and incubated overnight at room temperature in a humid chamber. Subsequently, the slides were washed with PBS (3 × 10 min), blocked in 10% v/v normal goat serum (Sigma) in blocking buffer (supernatant of 5% w/v milk powder in PBS centrifuged at maximum speed for 10 min) followed by staining with secondary antibody which was diluted in 10% v/v normal goat serum (Sigma) in blocking buffer and incubated for 2 h at room temperature in a humid chamber. Finally, the slides were washed with PBS (3 × 10 min) and embedded in Prolong Gold with DAPI (Invitrogen). The following primary antibodies were used: mouse anti-DMC1 (1:1000,Abcam ab11054), mouse anti-SYCP3 (1:200, Abcam ab97672), mouse anti-MLH1 (1:25, BD Pharmingen 51-1327GR), and rabbit polyclonal anti-RAD51 (1:1000)[46], rabbit anti-HSF2BP (1:30, #1)[22] and rabbit anti- BMRE1 (1:100, #2)[22], rabbit polyclonal anti-SYCP3(1:5000)[58] and guinea pig anti-SYCP2 (1:100)[59] and guinea pig anti-HORMAD2 (1:100)[60]. Secondary antibodies: goat anti-guinea pig Alexa 546 (Invitrogen, A-11074 1:500), goat anti-rabbit Alexa 488 (Invitrogen, A-11008 1:500), goat anti-rabbit Alexa 546 (Invitrogen, A-11010 1:500), goat anti-mouse Alexa 488 (Invitrogen, A-11001 1:500), goat anti-mouse Alexa 555 (Invitrogen, A-21422 1:500), and goat anti-mouse Alexa 633 (Invitrogen, A-21050 1:500).

Immunostained spreads were imaged using a Zeiss Confocal Laser Scanning Microscope 700 with 63x objective immersed in oil. All images within one analysis were taken with the same intensity. Images were analyzed using ImageJ (Fiji) software. BRME1, RAD51, and DMC1 foci quantification was performed using the ImageJ function "Analyze particles" in combination with a manual threshold and particles smaller than $0.0196\ \mu m^2$ and larger than $0.98\ \mu m^2$ were excluded. MLH1 foci were counted manually and blind by three individual researchers. HSF2BP foci quantification was performed using the ImageJ function "Analyze particles" in combination with a manual threshold. Since HSF2BP intensity is variable between slides and this influences the foci count because foci tend to merge as the intensity increases, size of particles was taken into account by adjusting the foci count in each nucleus in such a way that all large (equal to or larger than twice the average) HSF2BP-positive areas were divided by the average area size of $0.25\ \mu m^2$ to obtain more accurate foci number (this average focus size was calculated from particles analyzed of 15 wild-type and 16 $Brca2^{\Delta12/\Delta12}$ nuclei with various HSF2BP intensities). Particles smaller than $0.0294\ \mu m^2$ were excluded. For both BRME1 and HSF2BP a mask of SYCP3 was used to reduce the background signal.

**Statistical analysis.** Statistical analysis was performed using GraphPad Prism software version 9. Statistical significance was determined using one-way ANOVA with Tukey test for comparisons involving more than two groups (genotypes, treatments) or using unpaired two-tailed $t$-test for comparison between $+/+$ and $\Delta12/\Delta12$ only in the experiments where the $\Delta12/+$ group was not equally sampled ($n = 2$). Statistical details of the experiments can be found in the figure, figure legend or in the text of the results section, where $n$ represents the number of animals. In addition, the number of measurements (nuclei) is reported on the panels in Fig. 6. Exact $p$ values are reported on the panels.

**Animals.** All animals were kept in accordance with local regulations under the work protocols 17-867-11 and 15-247-20. Animal experiments were approved by the Dutch competent authority (Centrale Commissie Dierproeven) and all experiments conform to relevant regulatory standards. Female mice for CRISPR/Cas9 injection were C57BL/6 OlaHsd from Envigo, age 5 weeks. For spermatogenesis analysis male mice were sacrificed at the age of 6–15 weeks, except one wild-type (29 weeks) mouse used for RAD51, DMC1, and BRME1 foci quantification.

**Brca2-Δ12 mouse generation.** Brca2 Δ12 mice were generated by two CRISPR/Cas9 cut excision, as described before[16]. Female donor mice (age 5 weeks, C57BL/6 OlaHsd from Envigo) were superovulated by injecting 5–7.5 IE folligonan (100–150 μl), IP (FSH hormone; time of injection ± 13.30 h; day −3). Followed at day −1 by an injection of 5–7.5 IE chorulon (100–150 μl), IP (hCG hormone; time of injection 12.00 h). Immediately after the chorulon injection, the females were put with fertile males in a one to one ratio. Next day (0) females were euthanized by cervical dislocation. Oviducts were isolated, oocytes collected, and injected with ribonucleoprotein complexes of S.p.Cas9 3NLS (IDT cat. no. 1074181), crRNA, and tracrRNA (both Alt-R, synthesized by IDT). Target sequences for crRNA were TAATATTCCAACCCTCGTGT (upstream of Brca2 exon 12) and TGA-GAAATGTACACCTCATT (downstream of exon 12). For ribonucleoprotein formation equal volumes (5 μL) of crRNA and tracrRNA (both 100 μM in IDT annealing buffer) were mixed, heated to 95 °C for 5 min and allowed to cool on the bench. The annealed RNAs (1.2 μL, 50 μM) were mixed with Cas9 (10 μl diluted to 200 ng/μl in the DNA microinjection buffer (10 mM Tris-HCl, pH 7.4, 0.25 mM EDTA in water) at the final concentrations 0.12 μM Cas9, 0.6 μM of each of the two

crRNA:tracRNA complexes in microinjection buffer. Foster mothers (minimum age 8 weeks) were set up with vasectomized males in a 2–1 ratio. Next day (0), pseudopregnant female (recognized by a copulation prop) were collected. For transplanting the injected oocytes, pseudopregnant female was anesthetized by an IP injection of a mix of Ketalin (12 mg/ml ketamine in PBS)-Rompun (0.61 xylazine mg/ml PBS) 100 μl per 10 g bodyweight). Post-surgery pain relief was given when the mouse was anaesthetized (S.C. Rimadyl Cattle, 5 mg/ml in PBS, dose 5 μg/g mouse). Transplantation resulted in eight pups from a single litters, of which three (all female) contained the deletions in the targeted region, as determined by PCR genotyping. Different primer combinations were used for initial genotyping, but mB2i11-F1 AGCTGCCACATGGATTCTGAG, mB2i12-R2 GGACTAAGAGGCAAGGCATCA, and mB2e12-R1 GCTTTTTGAAGGTGT-TAAGGATTTT, were used routinely (Supplementary Table 1). Sequencing of the PCR products from the founder animals revealed mosaicism for the junctions between the two CRISPR/Cas9 cuts; deletion sizes were close to expected 713 bp, bigger (1179 bp) or smaller due to insertions of ectopic DNA. The experimental cohort was eventually formed through back-crossing and inter-crossing from one founder, with the deletion produced by direct ligation between the two Cas9 cuts. Routine PCR genotyping of was performed using MyTaq Red mix (Bioline) and using the mentioned primers in 1:0.5:1 combination, for simultaneous amplification of the wild-type and the Δ12 alleles (PCR products 663 and 314 bp, respectively). RT-PCR verification was performed on first-strand cDNA produced from testis RNA with oligo-dT primers and SuperScript II polymerase; the following primers were used: e11 ACATTTTCTGATGTTCCTGT; e12 GTGCCATCTGGA GTGCTTTT; e13 GTCGTGAGCCGGTAAGATTG; e14 TCCCTGGAGACACT CAGCTT; and e15 GAGCTGCTTAGGAGAACATGC.

Adult wild-type and $Brca2^{\Delta12}$ males were sacrificed and weighed, and testes and epididymides were collected and also weighed. Epididymides were collected in PBS, dounce homogenized, and sperm cells were counted. For histological analysis, testes were fixed in 4% PFA in PBS (overnight) and further processed for histological analysis using standard methods. Other testes were placed in PBS and further processed for immunocytochemistry as described in the corresponding section. For fertility assessment, breedings were set up between $Brca2^{\Delta12/\Delta12}$ and wild-type C57BL/6 animals.

**Immunofluorescence and microscopy.** Immunofluorescence staining was performed on ES cells grown overnight on a glass coverslip coated with laminin. Sterile 24 mm coverslip was placed in a 6-well plate, and a 100 μl drop of 0.05 mg/ml solution of laminin (Roche, 11243217001) was pipetted in the middle of it. The plate was left for ~30 min in the cell culture incubator, after which the laminin solution was aspirated, and cell suspension was placed in the well. DNA damage was induced by irradiation with 8 Gy X-ray followed by 2 h recovery. Cells were washed with PBS, pre-extracted in sucrose buffer (0.5% Triton X-100, 20 mM HEPES pH 7.9, 50 mM NaCl, 3 mM MgCl$_2$, 3 mM sucrose) for 1 min, fixed for 15 min in 2% paraformaldehyde in PBS at room temperature, immunostained with anti-RAD51 antibody and mounted with DAPI. Images were acquired using Leica SP8 confocal microscope in automatic tile scan mode. Maximum projections from a z-stack of three confocal planes through a 1 μm slice were produced for analysis. HSF2BP-GFP foci were quantified automatically using CellProfiler[61]. In short, nuclei were segmented using a global threshold (minimum cross-entropy) based on the DAPI signal. The masked images were used to identify HSFP2BP foci using a global threshold (Robust background) method with two standard deviations above background. Subsequently the number of foci was counted per segmented nucleus.

For single particle tracking, cells were grown overnight in eight-well glass bottom dishes (Ibidi) coated with 0.05 mg/ml laminin. Prior to the experiment cell medium was replaced with imaging medium (Fluorobrite DMEM (Thermo Fisher), complemented with 10% FCS, 1x NEAA, 89 μM β-mercaptoethanol, 200 U/ml penicillin, 200 μg/ml streptomycin, and 1000 U/ml leukemia inhibitory factor). Live-cell experiment was performed on a Zeiss Elyra PS complemented with a temperature-controlled stage and objective heating (TokaiHit). Samples were kept at 37 °C and 5% CO$_2$ while imaging. For excitation of GFP a 100 mW 488 nm laser was used. The samples were illuminated with HiLo illumination by using a 100×1.57NA Korr αPlan Apochromat (Zeiss) TIRF objective. Andor iXon DU897 was used for detection of the fluorescence signal, from the chip a region of 256 by 256 pixels (with an effective pixel size of 100 × 100 nm) was recorded at 19.2 Hz interval (50 ms integration time plus 2 ms image transfer time). EMCCD gain was set at 300. Per cell a total of 200 frames were recorded.

**Reporting summary.** Further information on research design is available in the Nature Research Reporting Summary linked to this article.

## Data availability
The coordinates and structure factors file for the HSF2BP-BRCA2 complex described in the study were deposited in Protein Data Bank[62], entry code 7BDX [https://doi.org/10.2210/pdb7bdx/pdb]. Source data are provided with this paper. The microscopy data generated in this study have been deposited in BioImage Archive[63] in BioStudies[64] database under accession code S-BIAD166 [https://www.ebi.ac.uk/biostudies/BioImages/studies/S-BIAD166]. Source data are provided with this paper.

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

## Acknowledgements

We thank Ambre Petitalot for the first purification of the BRCA2 F0 fragment, Christophe Velours on the I2BC protein–protein interaction facility for the SEC-MALS experiments, Guillaume Hoffmann and Jose A. Marquez for their advices during crystallogenesis assays on the HTX platform, and Virginie Ropars for her great help during crystallogenesis assays, crystal freezing, and X-ray crystallography data collection. We acknowledge SOLEIL for the provision of synchrotron radiation facilities and we would like to thank the respective staffs for assistance in using PROXIMA-1, PROXIMA-2, and SWING beamlines. We thank Nicole van Vliet for the assistance with the initial mouse experiments and Prof. Hyunsook Lee (Dept. of Biological Sciences, Seoul National University) for generously sharing the anti-BRCA2 antibody. We thank the Josephine Nefkens Cancer Progam for infrastructure support. The research leading to these results has received funding from the European Community's Seventh Framework Program H2020 under iNEXT (grant agreement N°653706). It was also supported by the French Infrastructure for Integrated Structural Biology (https://www.structuralbiology.eu/networks/frisbi, ANR-10-INSB-05-01), by the CNRS IR-RMN-THC Fr3050 and by the CEA. This research was also funded by the Dutch Cancer Society and by the Gravitation program CancerGenomiCs.nl from the Netherlands Organization for Scientific Research (NWO) and is part of the Oncode Institute, which is partly financed by the Dutch Cancer Society.

## Author contributions

R.K., S.Z.J. and A.N.Z. conceived the study. R.G., S.M., L.K., J.V., M.W.P., E.S.L., S.E.v.R.F. and A.N.Z. performed the experiments. M.H.L.D. and P.L. contributed to the determi-nation of the crystal structure. Y.v.L. and A.M. created and maintained the mouse lines. N.F.M. and A.M.P. generated reagents. J.E., W.M.B., R.K., S.Z.J. and A.N.Z. supervised the work. J.E., W.M.B., R.K. and S.Z.J. secured the funding. S.J.D., A.N.Z. and R.K. wrote the paper with contributions from the other authors.

## Competing interests

The authors declare no competing interests.
