## [Peer Review File · Nature Communications]

REVIEWER COMMENTS

Reviewer #1 (Remarks to the Author):

This article presents data to show that HSF2BP binds to BRCA2 with nanomolar affinities, and the interactions involve a BRCA2 peptide that induces tetramerization of ARM domains of HSF2BP. They further solved the crystal structure of 2 BRCA2 peptides bound to a tetramer of HSF2BP ARM domains to 2.6 Å. Using mutagenesis of residues involved in the interactions, they show that this binding site is required for the HSF2BP-BRCA2 interaction *in vivo*. However, they showed that this interaction is not required for the recruitment of BRCA2 to meiotic DSBs, which leaves the reader wondering about the relevance of the highly conserved BRCA2-HSF2BP interaction.

Furthermore, Zhang et al, 2020 (<https://www.nature.com/articles/s41467-020-15954-x>) have already shown that a mutant HSF2BP (which they call MEILB2), which is unable to bind BRCA2, can still localise to meiotic DSBs. They proposed that this is due to alternative recruitment by the ssDNA binding complex SPATA22-MEIOB. Similarly, the BRCA2 dependent tetramerisation of HSF2BP ARM domain dimers was already described in the Zhang et al 2020 paper.

So the novel new information presented here is the detailed interactions revealed by the crystal structure. However, it is questionable whether some or all the interactions are physiologically important/relevant.

- 1) To validate the interaction surfaces, the authors mutated large number of residues and assessed their interactions, mainly using co-IP. There are no quality measures to ensure that mutations have not altered the stability, structure and integrity of the proteins. It is therefore difficult to conclude that the lack of interactions is due to the loss of specific interactions or a gross change of the structure. The effects of single mutations in abolishing this high affinity interactions are surprising, especially when the deletion of exon 12 (the entire Motif 1) maintains weak interactions.
- 2) Figure 4 illustrated detailed mutagenesis that correlate with co-IP data from Figure 1. Indeed, the co-IP data do not completely support the interactions observed in the structure.
- 3) Since the interactions induce tetramerization, the ITC measurement and model need to take into consideration of the two distinct effects. How are the data modelled ?
- 4) Are the tetramerization physiologically relevant ? In addition to reduce interactions between BRCA2 and HSF2BP, deletion of exon12 also abolishes tetramerization.
- 5) Since the deletion of exon12 still allows interactions between BRCA2 and HSF2BP, the authors should conduct their *in vivo* experiments using deletion of exon 12-14 to assess the effects of abolished interactions.
- 6) The authors should further investigate their hypothesis that BRCA2 recruits HSF2BP to meiotic DSBs. For example, BRCA2 and SPATA22-MEIOB may be redundant pathways for recruiting HSF2BP and it would be interesting to test if the BRCA2 exon12-14-HSF2BP interaction becomes essential for HSF2BP recruitment in the MeioB^{-/-} mutant background.

Other points:

- 1) The colour scheme of the figures need to be improved. For example, orange and wheat (Figure 2) are similar colours and it is very difficult to distinguish BRCA2 peptide from ARM domain.
- 2) The figures need better annotations. For example, Figure 3C: It would be helpful to annotate which chain is which in the figure. The electrostatic potential surface is in a slightly different orientation compared to the surface in the panel above, which is confusing.
- 3) The detailed description of the structures is largely uninformative. The authors should simplify those sections that correspond to Figures 2-4.
- 4) The text in places are unclear and ambiguous. For example, in the abstract "This revealed two previously unrecognized BRCA2 repeats that each interact with one ARM monomer from two different

dimers."

5) The Sato 2020 reference on page 8 line 33 is in the wrong format.

6) There is no fully wild type control in figure 6k, it is unclear what the status of Brca2 is in the Hsf2bp +/+ data. When comparing the Brca2 $\Delta 12/+$ with the Brca2 $\Delta 12/\Delta 12$ there is a significant increase in the intensity of HSF2BP, but the implication isn't discussed.

Reviewer #2 (Remarks to the Author):

BRCA2 plays an important role in genome stability and interacts with multiple partners. Recently HSF2BP/MEILB2, a previously described testis specific protein has been found over expressed in tumor cell lines and identified as a BRCA2 interacting factor. HSF2BP is essential to the proper loading of the RAD51 recombinase and its meiotic specific paralog DMC1 during meiotic recombination but its exact role remains to be characterized. For these reasons, a good understanding of the role of BRCA2 and HSF2BP during meiotic recombination is essential to our knowledge of the maintenance of genetic integrity.

In previous works, the authors and other identified the BRCA2 and HSF2BP domains involved in their interaction. They also observed that this interaction increases the sensitivity of tumors cells to interstrand crosslinking agents. Another work strongly suggested that a BRCA2-HSF2BP interaction favors the recruitment of BRCA2 to meiotic chromosomes to ensure the loading of the recombinases. The aim of this work is to characterize the structural properties and functional consequences of the BRCA2-HSF2BP interaction. The authors solved the crystal structure of the complex between the involved domains and provided a dip and robust characterization of the interaction. The structural results are strongly sustained by Co-immunoprecipitation of mutated forms of the proteins expressed in two human cell lines. This part of the paper carefully highlights the structural role of a cryptic domain of BRCA2 that was not previously described. The second part of the paper attempts to determine the functional role of this interaction during meiotic recombination. The authors produced a transgenic mouse lacking the exo12 of the Brca2 gene. Given the results obtained in the first part of the paper this mutation is expected to strongly abolish the BRCA2-HSF2BP interaction and to induce a phenotype similar to the one of the Hsf2bp null mutant. Surprisingly the mutant does not phenocopy the Hsf2bp null mutant as male mutant mice complete meiosis and are fertile. Based on their results, the authors conclude that the deletion does not affect meiotic progression. However, the current version of the results needs to be improved to reach this conclusion. Appropriate statistical analysis are needed to analyze the graphs and an evaluation of the interaction of BRCA2 and HSF2BP remaining (or not) in the mutant testis would be highly informative to allow to formally conclude on its role.

- minor points

p6 : lane 11-13, the sentence is confusing, please clarify. Referring to figure 2D could help.

p33 : sup S1(C) "as panel A" should be "B"

Suggestions to help to read the figure

Add in A and D "transient HEK293T" like in B

Add in B and C "GFP-HSF2BP" beside the mutations

-major points

Fig 5 : G) As HSF2BP partially co-localize with RAD51, it would be of interest to analyze RAD51 behavior (foci number and intensity) in presence of the different BRCA2 constructs. From the immunostaining presented in the figure, it seems that RAD51 foci intensity is reduced in $\Delta 12-14$. Given the fact that HSF2BP is involved in RAD51 loading/stabilizing during meiotic recombination, this should represent a valuable information to better understand its role.

Fig 6 : This figure represents an important point of the paper as HSF2BP is essential to proper

prophase I completion in spermatocytes and previous results strongly suggested that HSF2BP meiotic activity involves its interaction with BRCA2. The author themselves ended their previous paper suggesting that "the highly conserved region of BRCA2 encoded by exon 12 and 13 has a function in meiosis rather than tumor suppression" (Brandsma et al. Cell Report 2019). Consequently, it is surprising that the suppression of exon 12, which strongly reduces HSF2BP/BRCA2 interaction in mitotic cells, has no impact on meiotic progression. For these reasons, this point has to be carefully clarified and the straightness of the results needs to be reinforced.

-The authors conclude that there are no differences between wt and mutants in graphs D, H, J and K without statistical results although variations are observed on graphs. When the number of mice is not sufficient to perform statistical analysis, please increase the sample number.

-BRME1 and HSF2BP foci numbers strongly decrease between zygotene and pachytene stages (Takemoto et al, Cell Reports 2020; Zhang et al, Nat Com 2019); to analyze differences between wt and mutants foci numbers should be scored independently in each stage.

-It is unclear why the authors choose to quantify "HSF2BP intensity" rather than counting foci numbers that is more accurate.

-The interaction between HSF2BP and BRCA2 in testis should be compared by Co-IP between wild type and mutant to conclude whether the interaction between these proteins is not essential to meiotic progression.

Reviewer #3 (Remarks to the Author):

At the editor's request, this review focuses on the NMR experiments described in the manuscript. The authors used a standard suite of triple resonance NMR experiments to assign HN, N, Ca, Cb, and CO resonances of a BRCA2 peptide. The assignments were used to estimate structural propensity of the peptide and to map regions of the BRCA2 peptide that interact with HSF2BP. Using Mulder et. al.'s ncSCP calculator, the authors determine that while the BRCA2 peptide is largely disordered in solution, residues 2291-2303 populate alpha helix at ~ 25 %. Other regions of the peptide also appear to populate helix albeit at reduced levels compared to 2291-2303. The authors conclude that "Only region N2291-S2303 forms a transient α -helix in more than 25 % of the molecules.". The way this statement is phrased leaves the reader with the impression that there is a mixed population of peptides: 25 % that form helix and 75 % that don't. This is not what the random coil-corrected chemical shifts indicate so the authors may want to consider re-phrasing this statement. The authors compare ¹H-¹⁵N HSQC cross peak intensities of ¹⁵N-labeled BRCA2 peptide in the presence and absence of unlabeled HSF2BP ARM domain to map the regions of the BRCA2 peptide that interact with the ARM domain. The authors observe a reduction in cross peak intensity of ~ 50 – 60 % across the BRCA2 peptide sequence. A signal intensity reduction of ~ 60 % appears to be specific to residues 2252-2342 of the peptide. From the spectrum, it appears that there are no changes to BRCA2 peptide cross peak positions. Based on ITC data, the authors conclude that a BRCA2 2252-2342 peptide binds to the ARM domain with ~ 10 nM affinity. The affinity measurement and the BRCA2 2252-2342 HSQC spectra +/- ARM domain seem at odds. Specifically, if the affinity of the complex were high, one would anticipate stable complex formation under the conditions of the NMR experiment (100 μ M peptide and 100 μ M ARM domain) and substantial changes to the BRCA2 2252-2342 cross peak positions. The authors must comment on why no changes in BRCA2 2252-2342 cross peak positions are observed upon addition of ARM domain.

Reviewer #4 (Remarks to the Author):

Ghouil et al.

The manuscript by Ghouil et al. describes the structural and functional studies of the BRCA2-HSF2BP interactions. A 2.6 Å crystal structure of the HSF2BP fragment in complex with two small BRCA2 fragments have been solved as a tetrameric complex with two different interfaces. Deletion of exon 12, containing the first interaction stretch, abolishes BRCA2 binding to HSF2BP in cells and shows defects in IR-induced co-localization of HSF2BP-GFP with RAD51. However, *Brca2*Δ12/Δ12 mice were fertile and did not show any meiotic defects, arguing against the functional importance of a HSF2BP-BRCA2 interaction. This is an important contribution that dissects the BRCA2-HSF2BP interaction and disproves the meiotic localizer model for HSF2BP. The identification of two structural repeats on BRCA2 that engage in the HSF2BP interaction adds significantly to our understanding of the large and complex BRCA2 protein. In a revision the authors should address the following points by additional clarification and discussion.

1. The size of the tetramer is about 76 Å, and the distance between the two F15X is presumably less than 38 Å. A dimeric BRCA2 is about 250 Å in size with an anti-parallel conformation (Shahid et al. NSMB 2014), indicating a much bigger distance between the two HSF2BP binding sites in the BRCA2 dimer. How could HSF2BP tetramer fit two full-length BRCA2 molecules? Is the tetramer conformation specifically induced by the binding of a much smaller F15X fragment? As the authors stated, there are two types of interfaces between the ARM domains (Page 6, line 1-4). In summary, it is not certain that full-length BRCA2 would cause tetramerization of the dimeric ARM domain, as concluded by the author on page 7 line 6 and a caveat to this effect should be added.
2. Recent work showed that BRCA2 oligomerization is regulated by its interaction with DSS1 and DNA, suggesting that monomeric BRCA2 is the active form for RAD51 loading (Le et al. 2020 NAR 48, 7818). The authors postulate here that HSF2BP binding may lead to BRCA2 dimerization and have published previously that HSF2BP binding negatively regulates BRCA2 function. Together with the point made above, the authors should expand the discussion on the significance and potential function of their observation that HSF2BP fragments lead to complex including a dimer of BRCA2 fragments.
3. The small sizes of F0 and F5X may lead to artificially high affinity in the interaction with HSF2BP. What is the interaction between the two full-length HSF2BP and BRCA2 proteins? An ITC experiment like in Figure 1E may not be possible due to the limiting amounts of full-length BRCA2, but maybe a caveat could be added.
4. In Figure 1B, the diagram presented BRCA2 fragments F14, F16 and F17 and their ability to bind to HSF2BP. However, I could not find but may have missed the underlying data showing the interaction of fragments F14 and F17 with HSF2BP.
5. Figure 1C: There is an unexplained discrepancy with fragment F16. It shows no interaction with WT HSF2BP, but the same level of interaction with the H3 fragment (the ARM domain) of HSF2BP like fragments F15 and F9. This is unexpected and unexplained. Does fragment H3 expose a hydrophobic patch that leads to artificial binding? This discrepancy should be acknowledged and the potential impact on experiments involving fragment H3 should be discussed.
6. On page 11, line 16, the authors state that "analysis of low frequency human polymorphisms revealed that single amino acid substitutions in BRCA2 and HSF2BP can be sufficient to disrupt the interaction, which might be clinically relevant for patients with either fertility defects or cancers." However, there is no meiotic phenotype of BRCA2 Δ12 in mice studies. Why would a phenotype then be expected in humans?

7. Figure 5A: It would help the reader to have the color code for exons 12, 13 14 use more distinct colors.

8. Supplement Fig. S1A: Which variant of R200 was used?

9. Supplement Fig. S1B: In the upper GFP blot, should the arrow position of GFP-HSF2BP band be the upper band?

10. Supplement Fig. S1B, C: Can the authors comment on the differences in interaction of some HSF2BP variants (M235T, K245T and R277E) and BRCA2 between HEK and HeLa? Are their BRCA2 sequence polymorphisms between both cell lines?

11. A minor point in the text and the diagram: The DMC1-binding sites have been assigned to the BRC6-8 repeats (Martinez et al, 2016, PNAS 113(13)). Is the exon 14 region identified in ref. 13 relevant?

Response to Reviewers

We are grateful to all four Reviewers for showing interest in our work and taking time to thoroughly review our data and conclusions and for providing valuable suggestions on how to improve the manuscript. We performed additional experiments and analyses and modified the text to address them.

Our responses are indicated in the following text style:

| Response

In addition to the changes made to address the points raised by the Reviewers, we modified the text to meet journal formatting requirements and changed the order of words in the title to make it more clear.

Response to Reviewer 1

This article presents data to show that HSF2BP binds to BRCA2 with nanomolar affinities, and the interactions involve a BRCA2 peptide that induces tetramerization of ARM domains of HSF2BP. They further solved the crystal structure of 2 BRCA2 peptides bound to a tetramer of HSF2BP ARM domains to 2.6 Å. Using mutagenesis of residues involved in the interactions, they show that this binding site is required for the HSF2BP-BRCA2 interaction *in vivo*. However, they showed that this interaction is not required for the recruitment of BRCA2 to meiotic DSBs, which leaves the reader wondering about the relevance of the highly conserved BRCA2-HSF2BP interaction.

Furthermore, Zhang et al, 2020 (<https://www.nature.com/articles/s41467-020-15954-x>) have already shown that a mutant HSF2BP (which they call MEILB2), which is unable to bind BRCA2, can still localise to meiotic DSBs. They proposed that this is due to alternative recruitment by the ssDNA binding complex SPATA22-MEIOB. Similarly, the BRCA2 dependent tetramerisation of HSF2BP ARM domain dimers was already described in the Zhang et al 2020 paper.

So, the novel new information presented here is the detailed interactions revealed by the crystal structure. However, it is questionable whether some or all the interactions are physiologically important/relevant.

We are grateful to the reviewer for the careful reading of our manuscript and the suggestions to improve it, most of which we were able to address in the revised version.

Our findings on ARM tetramerization independently corroborate using human HSF2BP and BRCA2 some of the data previously reported for the mouse proteins by Zhang *et al.* rather than duplicate it. We also want to stress that in addition to solving the structure, our major goal and an important (though unexpected) novelty of the paper, was testing the hypothesis that HSF2BP-BRCA2 interaction localizes the recombinases to meiotic DSBs. This data is described in the main Figures 5 and 6 and is based on the newly created and thus not yet published *Brca2* mouse strain and panels of genetically engineered mES cell lines we generated for this purpose. The Zhang et al 2020 experiment with overexpression of the mutant HSF2BP in testis the Reviewer refers to does not overlap this part of our work. That experiment could test whether BRCA2 is the “meiotic localizer” of HSF2BP, while the original Zhang 2019 paper postulated that HSF2BP is the meiotic localizer of BRCA2, i.e. the opposite.

Additionally, the mouse model and the mES cell line panels will help to address unresolved questions about the role of this BRCA2 region in somatic cells and will be of use to the much broader community characterizing BRCA2 variants in cancer.

1) To validate the interaction surfaces, the authors mutated large number of residues and assessed their interactions, mainly using co-IP. There are no quality measures to ensure that mutations have not altered the stability, structure and integrity of the proteins. It is therefore difficult to conclude that the lack of interactions is due to the loss of specific interactions or a gross change of the structure. The effects of single mutations in abolishing this high affinity interactions are surprising, especially when the deletion of exon 12 (the entire Motif 1) maintains weak interactions.

We should have stated more clearly that our variant analysis predated the solution of the structure. In case of HSF2BP it was based on a homology model and the one variant (R200T) we previously identified by arbitrarily changing 8 conserved residues and reported in Brandsma et al. 2019. Due to the high quality of the homology model nearly all predicted structural neighbors of R200 turned out to be involved in the interaction with BRCA2, and the structure later explained why. We modified the text in the results (p4 lines 26-31, p7 lines 8-18) and the Fig. 4 title to make this more clear.

Reduced stability of the variants could indeed result in the failure to co-immunoprecipitate. Since we first discovered HSF2BP-BRCA2 interaction we have tested >40 truncation and single residue substitution

variants and occasionally ran into this problem. For example, our initial prediction of the coiled coil - ARM boundary at aa 144 was incorrect and resulted in unstable fragments. Similarly, a natural splice variant of human HSF2BP with an in-frame skipping of exons 7 and 8, and one of the point mutants we tested (L85R) early on resulted in an unstable recombinant GFP-HSF2BP. This is illustrated on the panels from the early 2018 version of our paper that was eventually excluded from the published versions (refs 15, 16):

Left panel: immunoblots from the co-immunoprecipitation experiment with Flag-BRCA2-C-terminal fragment (bait) and the full-length GFP-HSF2BP and its deletion variants lacking the N-terminal coiled coil (ΔCC) or the ARM (Δarm) domains, and the natural splice variant lacking exons 7 and 8 (Δ7,8) (prey). The CC-ARM boundary was positioned too close to the C-terminus and resulted in unstable proteins (note weak bands in input αGFP, which correlated with weak αGFP signal in bound IP:Flag). **Middle panel:** low expression levels observed by immunoblot in the left panel correspond to reduced GFP signal measured by FACS. **Right panel:** Example of a single substitution L85R greatly reducing the amount of GFP-HSF2BP that can bind to GFP-agarose beads used for co-immunoprecipitation due to misfolding/reduced stability.

Based on this experience and the following consideration we argue that there is no indication that compromised folding affected the variants for which results are presented in the manuscript.

- Reduced cellular concentration of a variant due to instability manifests in the reduced intensity of the band in the total cell lysate (“input”). Reduced solubility of the bait (GFP-HSF2BP) due to aggregation caused by misfolding can impair its binding to the beads (bound α-GFP). There is no indication that any of this happens, as can be judged from the corresponding immunoblot panels we reported.
- Folding problems mostly arise from the disruption of the hydrophobic core. Unlike our previous blind screens, the one in this paper was focused on charged and polar residues, which were predicted — and later shown by the structure — to be surface-exposed (now clearly stated in the results section p 4 line 27). The only exception is G199, which is buried and which we targeted for being adjacent to R200.
- The BRCA2 region we focused on is intrinsically disordered (Supplementary Fig. 3) and as such immune to folding problems.
- Whenever possible we based our choice of mutation on human SNPs, to add another physiological component to our experiments.
- Most (12/15) of the single amino acid substitutions abolishing the interactions are in HSF2BP, which interacts with both BRCA2 motifs.
- The effect of Motif 1 (exon 12) deletion and disrupting mutations progressively decreases from *in vitro*, where HSF2BP-BRCA2 interaction is tested directly and in isolation, to *in vivo* experiments, where other supporting interactions may be in play:

- The affinity of purified proteins in the biochemical experiment is reduced ~1000x fold when exon 12 is deleted (Fig. 1e, 5b)

- Similarly complete effects are observed in the “in vitro biochemistry” setting (HEK293T Co-IPs, Fig S1) where both proteins are highly overproduced and thus out of balance with any other potential stabilizing partners
- Detectable residual co-precipitation (~5%) was observed from mES cells where both HSF2BP and BRCA2 are at physiological concentration
- Even more residual co-precipitation is observed in testis (new panel Fig. 6c), where HSF2BP has a known physiological function and was reported to bind protein-coated ssDNA, which is also a known substrate of BRCA2.

2) Figure 4 illustrated detailed mutagenesis that correlate with co-IP data from Figure 1. Indeed, the co-IP data do not completely support the interactions observed in the structure.

As mentioned in response to point 1, the mutations were designed before the structure was solved and were based on a homology model and human SNP data. With this clarification we hope that the Reviewer agrees with us that the mutagenesis is remarkably consistent with the structural findings, the discrepancies discussed below are minor, and the textual modification and figure changes we made address the criticism sufficiently.

To further address this, we now indicate not only the mutated residue, but also the selected mutations in Fig. 4. Also, we modified our description of the relationship between the co-IP and structural data to better highlight the consistency between both sets of data (see Results section 4 on p7). In particular, on the ARM side, mutations of residues E201 and K245 do not decrease binding, as observed by co-IP. Residues E201 and K245 interact through an intramolecular salt-bridge, which is disrupted by the mutations. They are in front of either F2293 or D2312 and K2316 in BRCA2. However, the presence of intermolecular electrostatic interactions involving E201 and K245 is not clear from our 3D structure and not supported by the co-IP data. In BRCA2, mutations E2292G, F2293L and P2334L do not decrease binding. However, E2292 is only slightly buried at the interface and isn't involved in any intermolecular salt-bridge, so that a Gly residue can be tolerated at position 2292. F2293 is packed against E201 and K245 and a Leu can make similar contacts. P2334 is packed against W132 and G133 and a Leu residue can also make similar contacts. In Fig. 4, the critical roles of N192 and N239 interacting with the BRCA2 backbone was illustrated in panel C. We now inserted an additional panel D to highlight the importance of taking into account the types of mutations while analyzing the impact of those mutations. We focused on the mutation P to L, comparing the impacts of P2329L and P2334L as a function of the structural context.

Minor point: when examining Fig. 4 during the revision of our manuscript, we noticed a mistake: in Fig. 4C, P2329 is indicated twice. We changed the blue label “P2329” to a blue label “P2334”.

3) Since the interactions induce tetramerization, the ITC measurement and model need to take into consideration of the two distinct effects. How are the data modelled ?

Indeed, we measured by ITC the heat resulting from both HSF2BP/ARM binding to BRCA2 and HSF2BP/ARM dimer binding to another HSF2BP/ARM dimer. However, as no tetramerization is observed in the absence of BRCA2, the K_d values displayed in Table 1 correspond to the dissociation of the tetrameric HSF2BP/ARM bound to BRCA2 into two dimeric HSF2BP/ARM and two BRCA2. Moreover, as the surface buried between the two ARM dimers, described in Supplementary S5B, is small relatively to the surface buried between ARM and BRCA2 (350 Å² vs twice 2740 Å²), tetramerization should not majorly contribute to the measured thermodynamics parameters.

This is now detailed in the paragraph 6 of the discussion (end of p11 – p12).

4) Are the tetramerization physiologically relevant ? In addition to reduce interactions between BRCA2 and HSF2BP, deletion of exon12 also abolishes tetramerization.

The conservation of the repeated motif in exons 12 and 13 provides a strong argument that tetramerization is under selective pressure and has an important physiological function. This remains one of the main conclusions from the paper stated in the abstract, and we added a sentence highlighting the evolutionary argument to the discussion (p11, end of the second paragraph).

5) Since the deletion of exon12 still allows interactions between BRCA2 and HSF2BP, the authors should conduct their in vivo experiments using deletion of exon 12-14 to assess the effects of abolished interactions.

We agree with the reviewer that the residual binding of HSF2BP to BRCA2 Δ 12 does not allow to conclusively rule out *any* role for HSF2BP-BRCA2 interaction in meiotic HR. A larger deletion would address this uncertainty but creating and characterizing a new mouse strain will take more than a year, even with animal ethics permits in place. In addition, the suggested deletion will remove the DMC1-binding site described by Thorslund et al., which will complicate the interpretation of any results obtained.

6) The authors should further investigate their hypothesis that BRCA2 recruits HSF2BP to meiotic DSBs. For example, BRCA2 and SPATA22-MEIOB may be redundant pathways for recruiting HSF2BP and it would be interesting to test if the BRCA2 exon12-14-HSF2BP interaction becomes essential for HSF2BP recruitment in the MeioB^{-/-} mutant background.

While this also is an experiment we would like to do, creating and intercrossing a new mouse model constitutes an entirely new study.

Other points:

1) The colour scheme of the figures need to be improved. For example, orange and wheat (Figure 2) are similar colours and it is very difficult to distinguish BRCA2 peptide from ARM domain.

We agree that it was difficult to distinguish the BRCA2 peptides from the ARM tetramer in Fig. 2d: we decided to represent the two peptides as tubes to make them thicker and more visible. We also changed the annotations of Fig. 2c,d,e in bold for clarity, and marked the N- and C-termini of the BRCA2 peptides in Fig. 2c.

2) The figures need better annotations. For example, Figure 3C: It would be helpful to annotate which chain is which in the figure. The electrostatic potential surface is in a slightly different orientation compared to the surface in the panel above, which is confusing.

We added chain labels in Fig. 3c. Also, in order to avoid any misunderstanding on the orientation of the electrostatic surface representation of ARM D, we changed its orientation to fit to the upper view of ARM D and added the position of the BRCA2 peptide (in black).

3) The detailed description of the structures is largely uninformative. The authors should simplify those sections that correspond to Figures 2-4.

We appreciate that the textual description of the structural details does not make for an easy read. In the revised version we made an effort to shorten this section, limit the description to the key points essential to support the conclusions we make, and transfer some of the descriptive details into the Fig. 2d legends.

4) The text in places are unclear and ambiguous. For example, in the abstract "This revealed two previously unrecognized BRCA2 repeats that each interact with one ARM monomer from two different dimers."

We had to shorten the abstract in half to fit the 150 word limit, so this specific ambiguous phrase is gone. We made an effort to revise similar descriptions throughout the text. At some point we got used to talking about “individual monomers from distinct dimers forming a tetramer” and lost the sense of how awkward it can sound. Thank you for pointing this out.

5) The Sato 2020 reference on page 8 line 33 is in the wrong format

This has been corrected, thank you.

6) There is no fully wild type control in figure 6k, it is unclear what the status of Brca2 is in the Hsf2bp +/- data. When comparing the Brca2 $\Delta 12/+$ with the Brca2 $\Delta 12/\Delta 12$ there is a significant increase in the intensity of HSF2BP, but the implication isn't discussed.

We are sorry for the non-uniform representation of the genotypes in different experiments reported in the reviewed manuscript. This has now been addressed by consistent statistical analysis of an extended dataset. Additionally, for HSF2BP we report foci number rather than intensity, as suggested by reviewer 2. The reasons and implications are discussed in detail in our response to that specific point.

Response to Reviewer 2.

BRCA2 plays an important role in genome stability and interacts with multiple partners. Recently HSF2BP/MEILB2, a previously described testis specific protein has been found over expressed in tumor cell lines and identified as a BRCA2 interacting factor. HSF2BP is essential to the proper loading of the RAD51 recombinase and its meiotic specific paralogue DMC1 during meiotic recombination but its exact role remains to be characterized. For these reasons, a good understanding of the role of BRCA2 and HSF2BP during meiotic recombination is essential to our knowledge of the maintenance of genetic integrity.

In previous works, the authors and other identified the BRCA2 and HSF2BP domains involved in their interaction. They also observed that this interaction increases the sensitivity of tumor cells to interstrand crosslinking agents. Another work strongly suggested that a BRCA2-HSF2BP interaction favors the recruitment of BRCA2 to meiotic chromosomes to ensure the loading of the recombinases.

The aim of this work is to characterize the structural properties and functional consequences of the BRCA2-HSF2BP interaction. The authors solved the crystal structure of the complex between the involved domains and provided a dip and robust characterization of the interaction. The structural results are strongly sustained by Co-immunoprecipitation of mutated forms of the proteins expressed in two human cell lines. This part of the paper carefully highlights the structural role of a cryptic domain of BRCA2 that was not previously described. The second part of the paper attempts to determine the functional role of this interaction during meiotic recombination. The authors produced a transgenic mouse lacking the exo12 of the Brca2 gene. Given the results obtained in the first part of the paper this mutation is expected to strongly abolish the BRCA2-HSF2BP interaction and to induce a phenotype similar to the one of the Hsf2bp null mutant. Surprisingly the mutant does not phenocopy the Hsf2bp null mutant as male mutant mice complete meiosis and are fertile. Based on their results, the authors conclude that the deletion does not affect meiotic progression. However, the current version of the results needs to be improved to reach this conclusion. Appropriate statistical analysis are needed to analyze the graphs and an evaluation of the interaction of BRCA2 and HSF2BP remaining (or not) in the mutant testis would be highly informative to allow to formally conclude on its role.

We want to thank the reviewer for the suggestions that allowed us to improve our study. The meiotic data we reported initially was indeed not always sufficiently uniform to draw conclusions supported by statistical analysis. We addressed this by adding new animals to ensure that there are at least three equally sampled animals in each experiment for the key comparison between +/+ and $\Delta 12/\Delta 12$ genotypes. Since we generally do not observe any differences in this comparison and to avoid sacrificing mice without good scientific justification, we did not add animals of the $\Delta 12/+$ genotype, but still kept the original IF data at $n=2$. We applied uniform statistical analysis to the expanded dataset, using one-way ANOVA with Tukey's test when three groups (genotypes) were present at $n \geq 3$, or t-test where only pairwise comparisons between +/+ and $\Delta 12/\Delta 12$ were possible. Exact p-values are now reported in Fig. 6, and the results and discussion sections have been modified to take them into account.

- minor points

p6 : lane 11-13, the sentence is confusing, please clarify. Referring to figure 2D could help.

Indeed, in the last lines of results section 2, referring to Fig. 2d now help the reader to go through our structural description of the HSF2BP-BRCA2 complex. Also, in the results section 3 lines 6-8, we modified the sentence to make it easier to follow the structural description of the interaction between the 4 HSF2BP monomers and the 2 BRCA2 peptide.

Suggestions to help to read the figure

Add in A and D "transient HEK293T" like in B

The label has been added.

Add in B and C "GFP-HSF2BP" beside the mutations

The label has been added.

-major points

Fig 5 : G) As HSF2BP partially co-localize with RAD51, it would be of interest to analyze RAD51 behavior (foci number and intensity) in presence of the different BRCA2 constructs. From the immunostaining presented in the figure, it seems that RAD51 foci intensity is reduced in $\Delta 12-14$. Given the fact that HSF2BP is involved in RAD51 loading/stabilizing during meiotic recombination, this should represent a valuable information to better understand its role.

We performed three additional experiments and quantified the RAD51 foci as well as HSF2BP-GFP foci. Imaging under uniform conditions, foci detection using automated software, and analysis of >200 nuclei per biological replica per condition, ensured that our new dataset is quantitative, robust and unbiased. The results reported in Fig. 5h did not reveal any significant differences in RAD51 foci numbers between cells expressing full-length and exon-deletion variants of *Brca2*. We replaced the images in Fig. 5g to represent the new dataset.

We could also compare integral intensities of the RAD51 foci in the new dataset. Although there is a small decrease in the Δ s, the difference apparent in the images we picked for the original paper, which were from irradiated cells, is small and not statistically significant (wt+IR vs $\Delta 12-14$ +IR, 7% reduction $p=0.4537$), although the difference between unirradiated cells of the same genotype is larger and significant (24% reduction, $p=0.0156$).

It is difficult to interpret these differences observed in somatic mES cells in the context of the role of HSF2BP-BRCA2 interaction, because complete HSF2BP knock-out has a very small effect in these cells, and is not associated with HR impairment (ref 16). Additionally, deletion of *Brca2* exons 12-14 goes beyond HSF2BP-binding domain and may affect other protein-protein interactions or have collateral effect on the adjacent (and important for somatic HR) DNA-binding domain of the BRCA2, complicating the interpretation. The purpose of these cell lines in the context of our paper was to test whether HSF2BP becomes functionally disengaged from BRCA2 $\Delta 12$. We believe the data we present meets this goal, and the hints of other effects are not strong enough and would only distract from the main message.

Fig 6 : This figure represents an important point of the paper as HSF2BP is essential to proper prophase I completion in spermatocytes and previous results strongly suggested that HSF2BP meiotic activity involves its interaction with BRCA2. The author themselves ended their previous paper suggesting that "the highly conserved region of BRCA2 encoded by exon 12 and 13 has a function in meiosis rather than tumor suppression" (Brandsma et al. Cell Report 2019). Consequently, it is surprising that the suppression of exon 12, which strongly reduces HSF2BP/BRAC2 interaction in mitotic cells, has no impact on meiotic progression. For these reasons, this point has to be carefully clarified and the straightness of the results needs to be reinforced.

Indeed, lack of phenotype was a complete surprise to us, both in the light of our prior expectation, and the combination of structural, evolutionary and biophysical data. We now performed more extensive comparison, statistical analysis and the testis co-IP experiment suggested by the Reviewer. They revealed some difference between *Brca2*^{+/+} and *Brca2* ^{$\Delta 12/\Delta 12$} animals. The $\Delta 12/\Delta 12$ have significantly heavier testes

than +/- control animals – opposite to the phenocopy of the reduced testis size of *Hsf2bp* ko strain we expected. Increasing the number of animals and quantifying foci in different prophase stages separately revealed small differences in BRME1, RAD51 and DMC1 foci, some of which are borderline-significant (results, last section p9, discussion end of p10 - p11).

The authors conclude that there are no differences between wt and mutants in graphs D, H, J and K without statistical results although variations are observed on graphs. When the number of mice is not sufficient to perform statistical analysis, please increase the sample number.

We regret insufficient and non-uniform sampling in our initial manuscript. As described above, this has now been addressed. Sample sizes are reported visually on the plots in Fig. 6 and consistent statistical analysis was performed.

BRME1 and HSF2BP foci numbers strongly decrease between zygotene and pachytene stages (Takemoto et al, Cell Reports 2020; Zhang et al, Nat Com 2019); to analyze differences between wt and mutants foci numbers should be scored independently in each stage.

We analysed foci in zygotene and pachytene separately as suggested, and in case of BRME1 also leptotene. HSF2BP foci are not detectable in leptotene in our hands. The data is reported in Fig. 6m,n.

It is unclear why the authors choose to quantify "HSF2BP intensity" rather than counting foci numbers that is more accurate.

To avoid bias we perform foci quantification using software. The pattern of HSF2BP staining we get with four different rabbit polyclonal antibodies in addition to focal had a pronounced axial element (both of which are absent in the *Hsf2bp*^{-/-}). The significance of this was not clear, so we initially chose to report the intensity of α -HSF2BP staining associated with meiotic chromosomes rather than the number of maxima (foci). Difference in the timing of the staining between HSF2BP (appears in zygotene) on the one hand, and BRME1, RAD51 and DMC1 on the other (detectable in leptotene), as well as the unexpected phenotype of our model, also contributed to our cautiousness.

While we still cannot explain the more extensive staining we observe with the available, knock-out-validated HSF2BP antibodies, the focal pattern is clearly present and can be quantified with some adjustments using software. We now report this measure for consistency with the other data.

The interaction between HSF2BP and BRCA2 in testis should be compared by Co-IP between wild type and mutant to conclude whether the interaction between these proteins is not essential to meiotic progression.

This is an important experiment we have been trying to do for a while. Immunodetection of mouse BRCA2 is complicated by lack of good antibodies, so we had to rely on ES cell experiments where BRCA2 could be tagged by a GFP knock-in and immunoprecipitated. We finally succeeded in co-IPing endogenous BRCA2 from testis using anti-HSF2BP and anti-RAD51 antibodies. The results are reported in Fig. 6c and discussed in the results section.

Response to reviewer 3

At the editor's request, this review focuses on the NMR experiments described in the manuscript.

We thank the reviewer for the critical assessment of the NMR results we report and for the helpful suggestions.

The authors used a standard suite of triple resonance NMR experiments to assign HN, N, Ca, Cb, and CO resonances of a BRCA2 peptide. The assignments were used to estimate structural propensity of the peptide and to map regions of the BRCA2 peptide that interact with HSF2BP.

Using Mulder et. al.'s ncSCP calculator, the authors determine that while the BRCA2 peptide is largely disordered in solution, residues 2291-2303 populate alpha helix at ~ 25 %. Other regions of the peptide also appear to populate helix albeit at reduced levels compared to 2291-2303. The authors conclude that "Only region N2291-S2303 forms a transient α -helix in more than 25 % of the molecules.". The way this statement is phrased leaves the reader with the impression that there is a mixed population of peptides: 25 % that form helix and 75 % that don't. This is not what the random coil-corrected chemical shifts indicate so the authors may want to consider re-phrasing this statement.

We agree that our sentence was inaccurate. We changed and extended this sentence to "BRCA2 F0 only forms transient α -helices; in particular, region N2291-S2303 folds into an α -helix that is present at about 25%.", in order to be more accurate (results section 1, p4 lines 39-40)

The authors compare ^1H - ^{15}N HSQC cross peak intensities of ^{15}N -labeled BRCA2 peptide in the presence and absence of unlabeled HSF2BP ARM domain to map the regions of the BRCA2 peptide that interact with the ARM domain. The authors observe a reduction in cross peak intensity of ~ 50 – 60 % across the BRCA2 peptide sequence. A signal intensity reduction of ~ 60 % appears to be specific to residues 2252-2342 of the peptide. From the spectrum, it appears that there are no changes to BRCA2 peptide cross peak positions. Based on ITC data, the authors conclude that a BRCA2 2252-2342 peptide binds to the ARM domain with ~ 10 nM affinity. The affinity measurement and the BRCA2 2252-2342 HSQC spectra +/- ARM domain seem at odds. Specifically, if the affinity of the complex were high, one would anticipate stable complex formation under the conditions of the NMR experiment (100 μM peptide and 100 μM ARM domain) and substantial changes to the BRCA2 2252-2342 cross peak positions. The authors must comment on why no changes in BRCA2 2252-2342 cross peak positions are observed upon addition of ARM domain.

We agree that we did not explain this result. It comes from the stoichiometry of the binding reaction. We now added, after the description of the ITC results in results section 1, p5 lines 6-8): "These affinity and stoichiometry were consistent with the more than 2-fold decrease in intensity observed by NMR when adding one ARM to one ^{15}N labeled F0; indeed, in these conditions, half of the F0 molecules were free and half bound to ARM."

Response to Reviewer 4

The manuscript by Ghouil et al. describes the structural and functional studies of the BRCA2-HSF2BP interactions. A 2.6 Å crystal structure of the HSF2BP fragment in complex with two small BRCA2 fragments have been solved as a tetrameric complex with two different interfaces. Deletion of exon 12, containing the first interaction stretch, abolishes BRCA2 binding to HSF2BP in cells and shows defects in IR-induced co-localization of HSF2BP-GFP with RAD51. However, *Brca2* Δ 12/ Δ 12 mice were fertile and did not show any meiotic defects, arguing against the functional importance of a HSF2BP-BRCA2 interaction. This is an important contribution that dissects the BRCA2-HSF2BP interaction and disproves the meiotic localizer model for HSF2BP. The identification of two structural repeats on BRCA2 that engage in the HSF2BP interaction adds significantly to our understanding of the large and complex BRCA2 protein. In a revision the authors should address the following points by additional clarification and discussion

We are grateful to the reviewer for the thorough assessment of our work and for the valuable suggestions.

1. The size of the tetramer is about 76 Å, and the distance between the two F15X is presumably less than 38 Å. A dimeric BRCA2 is about 250 Å in size with an anti-parallel conformation (Shahid et al. NSMB 2014), indicating a much bigger distance between the two HSF2BP binding sites in the BRCA2 dimer. How could HSF2BP tetramer fit two full-length BRCA2 molecules? Is the tetramer conformation specifically induced by the binding of a much smaller F15X fragment? As the authors stated, there are two types of interfaces between the ARM domains (Page 6, line 1-4). In summary, it is not certain that full-length BRCA2 would cause tetramerization of the dimeric ARM domain, as concluded by the author on page 7 line 6 and a caveat to this effect should be added.

The discussion of our findings on BRCA2 and HSF2BP oligomerisation has been revised for clarity and accuracy (end of p11-p12). However, most of the argument the Reviewer makes here appears to stem from the difference in the view of the prior data on the structure and oligomeric state of BRCA2. Several points and an illustration of what we can deduce from our own data on HSF2BP (full-length and ARM) and BRCA2 hopefully can further clarify our vision:

- While published structural data clearly indicate that the folded core of BRCA2 is an antiparallel dimer, a large portion of BRCA2 contains intrinsically disordered regions allowing very dynamic conformational changes. Using techniques that do not depend on averaging BRCA2 particles, extended BRCA2 conformations and higher ordered oligomeric forms of BRCA2 have been revealed by *in vitro* experimnts (see for example scanning force microscopy data in Sanchez et al PMID: 28168276, Sidhu et al. PMID: 32785644); *in vivo* data also reveal larger than dimeric complexes (Reuter et al. PMID: 25488918). These

Top: Schematic of the full-length HSF2BP and the three oligomerization mechanisms discussed in the paper. **Middle:** Schematic representation and a structural model showing how CryoEM BRCA2 dimer structure can be oligomerized by HSF2BP ARM tetramers despite the difference in distances mentioned by the Reviewer. **Bottom:** A schematic of the hypothetical oligomerization of the BRCA2 multimers observed by scanning force microscopy by the full-length HSF2BP tetramers.

properties of BRCA2 can accommodate a broad range of BRCA2 interactions (all of which with the antiparallel dimer at its core). In the Figure presented here we illustrate how a HSF2BP tetramer can fit two full-length BRCA2 molecules.

- even assuming the rigid anti-parallel dimer model and a mismatch in distances the Reviewer points out, HSF2BP tetramer can interact with two BRCA2 dimers, as illustrated in the drawing
- The most solid consequence of this interaction is that BRCA2 concentration will increase locally, as stated in the paragraph 6 of the discussion (p12). Which could also modify the oligomerization state of BRCA2, shifting from a monomeric/dimeric state to a higher oligomeric state.
- In cells BRCA2 can oligomerise through other interacting partners, in particular PALB2 is key to BRCA2 oligomerisation and interaction with BRCA1; HSF2BP dimers can bridge BRCA2-PALB2 hetero-oligomers.

2. Recent work showed that BRCA2 oligomerization is regulated by its interaction with DSS1 and DNA, suggesting that monomeric BRCA2 is the active form for RAD51 loading (Le et al. 2020 NAR 48, 7818). The authors postulate here that HSF2BP binding may lead to BRCA2 dimerization and have published previously that HSF2BP binding negatively regulates BRCA2 function. Together with the point made above, the authors should expand the discussion on the significance and potential function of their observation that HSF2BP fragments lead to complex including a dimer of BRCA2 fragments.

Again, we only meant that HSF2BP binding locally concentrates BRCA2, and thus favours its oligomerization. However, we agree that, if monomeric BRCA2 is the active form for HR, then favouring BRCA2 oligomerization impairs BRCA2 function in HR.

3. The small sizes of F0 and F5X may lead to artificially high affinity in the interaction with HSF2BP. What is the interaction between the two full-length HSF2BP and BRCA2 proteins? An ITC experiment like in Figure 1E may not be possible due to the limiting amounts of full-length BRCA2, but maybe a caveat could be added.

Yes, sadly it is totally impossible to measure by ITC the affinity between full-length HSF2BP and BRCA2 proteins, due to limited amount of BRCA2 purified from mammalian cells.

4. In Figure 1B, the diagram presented BRCA2 fragments F14, F16 and F17 and their ability to bind to HSF2BP. However, I could not find but may have missed the underlying data showing the interaction of fragments F14 and F17 with HSF2BP.

Thank you for pointing this out. We added a panel to Fig. S1 (S1B).

5. Figure 1C: There is an unexplained discrepancy with fragment F16. It shows no interaction with WT HSF2BP, but the same level of interaction with the H3 fragment (the ARM domain) of HSF2BP like fragments F15 and F9. This is unexpected and unexplained. Does fragment H3 expose a hydrophobic patch that leads to artificial binding? This discrepancy should be acknowledged and the potential impact on experiments involving fragment H3 should be discussed.

Again, thank you for pointing out this inconsistency, which we missed. WT HSF2BP – F16 did co-precipitate with reduced efficiency in the experiment shown in the new panel S1B, which we added in response to the previous point. The discrepancy is likely due to the low amount of GFP-HSF2BP bound to the beads (α -GFP bound) in Fig 1C compared to Fig S1B, which pushes the system beyond detectability threshold. F16 lacks part of motif 2.

6. On page 11, line 16, the authors state that “ analysis of low frequency human polymorphisms revealed that single amino acid substitutions in BRCA2 and HSF2BP can be sufficient to disrupt the interaction, which might

be clinically relevant for patients with either fertility defects or cancers.” However, there is no meiotic phenotype of BRCA2 Δ 12 in mice studies. Why would a phenotype then be expected in humans?

Yes, this sentence did not match our findings. We removed it.

7. Figure 5A: It would help the reader to have the color code for exons 12, 13 14 use more distinct colors.

We modified Fig. 5a to use distinct color codes, and consistently changed the color codes in Fig. 1d and Fig. 5c.

8. Supplement Fig. S1A: Which variant of R200 was used?

We adjusted the labelling to indicated that it was R200T.

9. Supplement Fig. S1B: In the upper GFP blot, should the arrow position of GFP-HSF2BP band be the upper band?

(now Supplementary Fig. 1c) The upper band. We adjusted the position of the arrow, thank you for pointing this out.

10. Supplement Fig. S1B, C: Can the authors comment on the differences in interaction of some HSF2BP variants (M235T, K245T and R277E) and BRCA2 between HEK and HeLa? Are their BRCA2 sequence polymorphisms between both cell lines?

(now Supplementary Fig. 1c,d) HEK293T cells have much higher expression levels due to highly efficient transient transfection, which are toxic/nonphysiological even if the produced proteins are intrinsically innocuous. In our experience these conditions are more likely to cause artifacts (co-aggregation of unfolded proteins stuck in ER ?) and capture weak interaction. Establishment of stable cell lines greatly reduces overexpression levels: only a limited number of copies of the transfected DNA remain integrated into chromosome, cells producing toxic amounts of proteins are overgrown by those which produce moderate amounts.

11. A minor point in the text and the diagram: The DMC1-binding sites have been assigned to the BRC6-8 repeats (Martinez et al, 2016, PNAS 113(13). Is the exon 14 region identified in ref. 13 relevant?

While there is some controversy regarding the exon 14 site, the data that challenges the relevance of the biochemical findings reported in ref 13 is limited (Biswas et al. 2012, PMID: 22678057), and in our view not sufficient to ignore the extensive in vitro characterisation by Thorslund and West. Just like the HSF2BP-binding region in exons 12-13, the PhePP motif in exon 14 is highly conserved and has no known function in somatic cells, so its involvement in DMC1 control in meiosis seem plausible. This, of course, does not mean that DMC1 binding to BRCA2 BRC repeats does not occur — both sites can co-exist.

REVIEWERS' COMMENTS

Reviewer #1 (Remarks to the Author):

The revised manuscript has addressed all the minor points I have raised and improved the clarity and readability of the manuscript. However the major concerns I raised: the physiological relevance of the tetramer and the physiological significance of the interactions, remain unanswered. The study provides a structural basis and molecular details of how a segment (~ 50 amino acids long peptide) of BRCA2 (3284 amino acids long) interact with a domain (ARM domain) in HSF2BP. When part or all of this BRCA2 segment is deleted, the interaction is significantly reduced in vitro (using peptides) and between BRCA2 and HSF2BP in vivo (as demonstrated by Co-IP and co-localisation), confirming that this interaction surface exists in vivo. However, there is a lack of clear phenotype when the peptide is deleted, especially when only part of the peptide is deleted, suggesting that the significance of these interactions remains to be determined.

In summary, this study provides the structural basis for an interaction site between BRCA2 and HSF2BP. However, the mechanistic and physiological implications of this interaction and indeed the importance of the association between BRCA2 and HSF2BP remain to be determined. It would have made the conclusions much less ambiguous if the effects (or lack of it) are demonstrated when both exon12 and exon13 (the full peptide) are deleted in mice.

Specific points:

- 1) the affinities involving peptide-induced tetramerisation might not represent the affinity between full length proteins. As I previously stated, the induced tetramerization of ARM domain upon binding to the peptides adds complications in measuring binding affinity (dissociation constant) and thus the low nM affinity is misleading. The real affinity can only be obtained by using full length BRCA2 and full length HSF2BP, which are unamenable due to the difficulty in obtaining the full length BRCA2 proteins. Despite the arguments presented in the manuscript by the authors about the potential evolutionary and functional relevance of the induced tetramerization, interactions and oligomerisations between domains/peptides are not always functionally relevant in full length proteins as demonstrated by many crystal packing-induced oligomerisations/interactions of protein domains in literature.
- 2) The lack of severe phenotype in cell lines and mutant mice testis when part of the peptide (exon 12) is deleted, which results in weaker (but not abolished) interactions between the peptide and ARM domain in vitro and between BRCA2 and HSF2BP in vivo, suggests that the weaker association might be sufficient for its functions in meiosis.
- 3) deleting the full peptide did not abolish the interactions in vivo (Figure 5d-f). The authors thus can't rule out the possibility that there might be other weak interaction sites or indirect interactions via other partners under normal physiological conditions to allow the association between BRCA2 and HSF2BP. It is unclear what the two bands corresponding to HSF2BP are in Figure 5d. The amount of HSF2BP is different in Figure 2e and thus can't compare the amount of BRCA2.
- 4) Combining points 2 and 3, the study overall does not rule out the importance of the association between BRCA2 and HSF2BP in meiosis, either through the identified site or via other sites/cofactors.

Reviewer #2 (Remarks to the Author):

The authors answered all questions and improved substantially the manuscript, in particular they provided a key element confirming that the deletion of exon12 of HSF2BP reduces its interaction with BRCA2 in meiocytes allowing to conclude that this interaction is not essential to meiotic progression. The more complete and robust analysis of the meiotic phenotype presented in the revised manuscript provides an important set of data to better understand the role of BRCA2 during meiosis.

Reviewer #3 (Remarks to the Author):

I had a couple of very minor concerns regarding the NMR data contained in the initially submitted manuscript. These concerns were addressed in full in the re-submitted manuscript.

Reviewer #4 (Remarks to the Author):

Ghouil et al. revised manuscript

The authors addressed all concerns by their clarifications and changes in the text and figures. There is one remaining minor issue with Figure S1c.

1. Supplement Fig. S1c: In the upper GFP blot, should the arrow position of GFP-HSF2BP band be the upper band?

The arrow position for GFP-HSF2BP in the upper blot (bound) should be adjusted to the upper band.

Response to Reviewer 1

The revised manuscript has addressed all the minor points I have raised and improved the clarity and readability of the manuscript. However, the major concerns I raised: the physiological relevance of the tetramer and the physiological significance of the interactions, remain unanswered. The study provides a structural basis and molecular details of how a segment (~ 50 amino acids long peptide) of BRCA2 (3284 amino acids long) interact with a domain (ARM domain) in HSF2BP. When part or all of this BRCA2 segment is deleted, the interaction is significantly reduced in vitro (using peptides) and between BRCA2 and HSF2BP in vivo (as demonstrated by Co-IP and co-localisation), confirming that this interaction surface exists in vivo. However, there is a lack of clear phenotype when the peptide is deleted, especially when only part of the peptide is deleted, suggesting that the significance of these interactions remains to be determined. In summary, this study provides the structural basis for an interaction site between BRCA2 and HSF2BP. However, the mechanistic and physiological implications of this interaction and indeed the importance of the association between BRCA2 and HSF2BP remain to be determined. It would have made the conclusions much less ambiguous if the effects (or lack of it) are demonstrated when both exon12 and exon13 (the full peptide) are deleted in mice.

Specific points:

- 1) the affinities involving peptide-induced tetramerisation might not represent the affinity between full length proteins. As I previously stated, the induced tetramerization of ARM domain upon binding to the peptides adds complications in measuring binding affinity (dissociation constant) and thus the low nM affinity is misleading. The real affinity can only be obtained by using full length BRCA2 and full length HSF2BP, which are unamenable due to the difficulty in obtaining the full length BRCA2 proteins. Despite the arguments presented in the manuscript by the authors about the potential evolutionary and functional relevance of the induced tetramerization, interactions and oligomerisations between domains/peptides are not always functionally relevant in full length proteins as demonstrated by many crystal packing-induced oligomerisations/interactions of protein domains in literature.
- 2) The lack of severe phenotype in cell lines and mutant mice testis when part of the peptide (exon 12) is deleted, which results in weaker (but not abolished) interactions between the peptide and ARM domain in vitro and between BRCA2 and HSF2BP in vivo, suggests that the weaker association might be sufficient for its functions in meiosis.
- 3) deleting the full peptide did not abolish the interactions in vivo (Figure 5d-f). The authors thus can't rule out the possibility that there might be other weak interaction sites or indirect interactions via other partners under normal physiological conditions to allow the association between BRCA2 and HSF2BP. It is unclear what the two bands corresponding to HSF2BP are in Figure 5d. The amount of HSF2BP is different in Figure 2e and thus can't compare the amount of BRCA2.
- 4) Combining points 2 and 3, the study overall does not rule out the importance of the association between BRCA2 and HSF2BP in meiosis, either through the identified site or via other sites/cofactors.

We agree with the key points made by the reviewer. Our work does not rule out that some HSF2BP-BRCA2 interaction is involved in meiosis — only that the one we described structurally isn't. New mouse models are required to address this issue — they just cannot be promptly produced. Absence of phenotypes in *Brca2*^{A12/A12} mice *does* raise the question about the physiological relevance of the oligomerisation and the repeat conservation — our study will hopefully become a starting point for further work to uncover it.

We were aware of these limitations throughout the study, strived to be open about them when writing the manuscript and were careful not to over-interpret our data when drawing conclusion. Among other things, because we have ongoing work that depends on getting this right. We further checked the text to make sure the caveats are clearly indicated:

- in the summary paragraph of the Introduction (p4) we specified that our findings demonstrate that “**evolutionarily conserved high-affinity oligomerisation-inducing**” interaction is not essential for meiotic HR, as indeed it is possible that weak, indirect interactions, whose signs are present in our in vivo data, are sufficient to compensate. This wording was already used in the other parts of the manuscript and in our view accurately reflects the conclusions that can be made.

- we modified the concluding paragraph of the Discussion (p12), to explicitly state that our findings do not rule out that HSF2BP and BRCA2 must co-localise at meiotic DSBs to facilitate HR
- we further expanded the description of the possible compensatory interactions that may mask the phenotype in our mouse model (last sentences of the Results section, p9)
- in the first paragraph of the Discussion (bottom of p9) we changed the text from “eliminating the HSF2BP-binding region” to “disrupting the HSF2BP-binding region” — since we did not eliminate the region completely

We are working on making new mouse models in which BRCA2-HSF2BP interaction will be completely abolished: from the BRCA2 side (exon 12-13 deletion) and/or from the HSF2BP side. As we mentioned in the previous response, this requires much more time than is allowed for the revision, and there are uncertainties involved.

Response to Reviewer 2

The authors answered all questions and improved substantially the manuscript, in particular they provided a key element confirming that the deletion of exon12 of HSF2BP reduces its interaction with BRCA2 in meiocytes allowing to conclude that this interaction is not essential to meiotic progression. The more complete and robust analysis of the meiotic phenotype presented in the revised manuscript provides an important set of data to better understand the role of BRCA2 during meiosis.

| We thank the reviewer for the helpful suggestions during the initial review and for assessing the revision.

Response to Reviewer 3

I had a couple of very minor concerns regarding the NMR data contained in the initially submitted manuscript. These concerns were addressed in full in the re-submitted manuscript.

We want to thank the reviewer again for helping us make the description of the NMR experiments more clear and consistent.

Response to Reviewer 4

The authors addressed all concerns by their clarifications and changes in the text and figures. There is one remaining minor issue with Figure S1c.

1. Supplement Fig. S1c: In the upper GFP blot, should the arrow position of GFP-HSF2BP band be the upper band?

The arrow position for GFP-HSF2BP in the upper blot (bound) should be adjusted to the upper band.

This has now been corrected. Thank you very much again for your helpful suggestions and taking a close, detailed and insightful look at our work!